# A General Method for Robust Learning from Batches

Ayush Jain and Alon Orlitsky
Dept. of Electrical and Computer Engineering
University of California, San Diego
{ayjain, aorlitsky}@eng.ucsd.edu

## Abstract

In many applications, data is collected in batches, some of which may be corrupt or even adversarial. Recent work derived optimal robust algorithms for estimating finite distributions in this setting. We develop a general framework of robust learning from batches, and determine the limits of both distribution estimation, and notably, classification, over arbitrary, including continuous, domains.

Building on this framework, we derive the first robust agnostic: (1) polynomial-time distribution estimation algorithms for structured distributions, including piecewise-polynomial, monotone, log-concave, and gaussian-mixtures, and also significantly improve their sample complexity; (2) classification algorithms, and also establish their near-optimal sample complexity; (3) computationally-efficient algorithms for the fundamental problem of interval-based classification that underlies nearly all natural-1-dimensional classification problems.

## 1 Introduction

### 1.1 Motivation

In many learning applications, some samples are inadvertently or maliciously corrupted. A simple and intuitive example shows that this erroneous data limits the extent to which a distribution can be learned, even with infinitely many samples. Consider $p$ that could be one of two possible binary distributions: $(\frac{1}{2} - \frac{\beta}{2}, \frac{1}{2} + \frac{\beta}{2})$ and $(\frac{1}{2} + \frac{\beta}{2}, \frac{1}{2} - \frac{\beta}{2})$. Given any number of samples from $p$, an adversary who observes a $1 - \beta$ fraction of the samples and can determine the rest, could use the observed samples to learn $p$, and set the remaining samples to make the distribution always appear to be $(0.5, 0.5)$. Even with arbitrarily many samples, any estimator for $p$ fails to decide which $p$ is in effect, hence incurs a *total-variation (TV)* distance $\geq \frac{\beta}{2}$, that we call the *adversarial lower bound*.

The example may seem to suggest the pessimistic conclusion that if an adversary can corrupt a $\beta$ fraction of the data, a TV-loss of $\geq \frac{\beta}{2}$ is inevitable. Fortunately, in many applications it can be avoided.

In the following applications, and many others, data is collected in batches, most of which are genuine, but some possibly corrupted. Data may be gathered by sensors, each providing a large amount of data, and some sensors may be faulty. The word frequency of an author may be estimated from several large texts, some of which are mis-attributed. User preferences may be learned by querying several individuals, some intentionally biasing their feedback. Multiple agents may contribute to a crowd-sourcing platform, but some may be unreliable or malicious. Interestingly, for data arriving in batches, even when a $\beta$-fraction of which are corrupted, more can be said.

Recently, [QV17] formalized the problem for finite domains. They considered estimating a distribution $p$ over $[k]$ in TV-distance when the samples are provided in batches of size $\geq n$. Out of a total of $m$ batches, a fraction $\leq \beta$ may be arbitrarily and adversarially corrupted, while in every other batch $b$ the samples are drawn according to a distribution $p$.

For $\beta < 1/900$, they derived an estimation algorithm that approximates any $p$ over a finite domain to TV-distance $\mathcal{O}(\beta/\sqrt{n})$, surprisingly, much lower than the individual samples limit of $\Theta(\beta)$. They also derived a matching lower bound, showing that even for binary distributions, and hence for general finite distributions, given any number $m$ of batches, the lowest achievable TV distance is $\Delta_{\min} := \Delta_{\min}(\beta, n) := \frac{\beta}{2\sqrt{2n}}$. We refer to $\Delta_{\min}$ as the *adversarial batch lower bound*.

Their estimator requires $\Omega(\frac{n+k}{n \cdot \Delta_{\min}^2})$ batches of samples, or equivalently $\Omega(\frac{n+k}{\Delta_{\min}^2})$ samples, which is not optimal if $n >> k$. It also runs in time exponential in the domain size, rendering it impractical.

Recently, [CLM19] used a novel application of the sum-of-squares technique to reduce the exponential time complexity. Using quasi-polynomial sample size and run time, both roughly $(k/\Delta)^{\mathcal{O}(\log(1/\beta))}$, they derived an estimator that achieves TV distance $\mathcal{O}(\Delta)$, where $\Delta := \Delta(\beta, n) := \Delta_{\min} \cdot \sqrt{\ln(1/\beta)}$.

Concurrently, [JO19] derived the first polynomial-time and optimal $\Omega(k/\Delta^2)$ sample estimator, that achieves the same $\mathcal{O}(\Delta)$ TV distance. To limit the impact of adversarial batches, the algorithm *filters* the data by removing batches that skews the estimator.

For general distributions, the sample complexity of both TV-distance estimation, and Bayes-optimal classification, grows linearly in the domain size, even when all samples are genuine. Hence, general estimation and classification over large discrete, let alone continuous domains, is infeasible. Since most modern applications are over very large or continuous domains, this may again lead to the pessimistic conclusion that not much can be done.

Fortunately, typical distributions are not arbitrary and possess some structure. For example, they may be monotone, smooth, Lipchitz, etc., or well approximated by structured distributions. These structural properties enable learning over large and even infinite domains. For example, as is well known, classifiers can be learned using a number of samples proportional to the VC-dimension of the classifier class. But so far, our understanding of how to incorporate the distribution structure in Robust batch learning has been quite limited.

The first application of structure to reduce the linear dependence of the sample complexity [CLM19] considered robust batch learning of $t$-piecewise degree-$d$ polynomials over the finite set $[k] = \{1, \ldots, k\}$. It learned these distributions with number of samples that grows only quasi-poly-logarithmically in the domain size $k$. Yet this number still grows with $k$, hence does not extend to continuous distributions. It is also quasi-polynomial in the other parameters $t$, $d$, batch size $n$, and $1/\beta$, much larger than in the non-robust setting. And the algorithm's computational complexity is quasi-polynomial in these parameters and the domain size $k$.

This leaves several natural questions: (1) Can other non-finite, and even continuous, structured distribution classes, be learned robustly to an estimation error comparable to the adversarial batch lower $\Delta_{\min}$? (2) Can it be achieved with sample complexity comparable to the non-adversarial learning? (3) Can robust learning of structured distributions be accomplished in strict polynomial time? (4) Even more generally can other tasks such as classification be accomplished with adversarial batches? (5) Most importantly, is there a general and systematic theory of learning with adversarial batches?

## 1.2 Summary of techniques and contributions

VC theory helps answer some of the above questions when all the samples are generated i.i.d. from a distribution. We adapt the theory to address robust batch learning as well. Let $\mathcal{F}$ be a family of subsets of an Euclidean domain $\Omega$. The $\mathcal{F}$-*distance* between two distributions $p$ and $q$ over $\Omega$ is the largest difference between the probabilities $p$ and $q$ assign to any subset in $\mathcal{F}$,

$$||p - q||_{\mathcal{F}} := \sup_{S \in \mathcal{F}} |p(S) - q(S)|.$$

It is easy to see that TV, and hence $L_1$, distances are a special case of $\mathcal{F}$-distance where $\mathcal{F}$ is the collection $\Sigma$ of all Borel subsets of $\Omega$, $||p - q||_{\Sigma} = ||p - q||_{\text{TV}} = \frac{1}{2}||p - q||_1$.

Without adversarial batches, the VC inequality guarantees that for a subset family $\mathcal{F}$ with finite VC-dimension, the empirical distribution of samples from $p$ estimates $p$ to a small $\mathcal{F}$-distance. But with adversarial batches, the $\mathcal{F}$-distance between the empirical distribution and $p$ could be large.

For learning with adversarial batches over finite domains, [JO19] presented an algorithm that learns the distribution to a small TV distance with a number of batches proportional to the domain size. We

generalize this algorithm to learn any finite-VC subset family $\mathcal{F}$ to a small $\mathcal{F}$-distance using samples linear in the family's VC-dimension, rather than the domain size.

Recall that $\Delta_{\min} = \beta/(2\sqrt{2n})$ is the adversarial batch lower bound for TV-distance learning. No algorithm achieves an error below $\Delta_{\min}$, even with the number of batches $\to \infty$. Since the $\Delta_{\min}$ lower bound applies even to binary domains, it can be shown to also lower bound $\mathcal{F}$-distance learning.

Our proposed algorithm *filters* the batches and returns a sub-collection of batches whose empirical distribution estimates $p$ to $\mathcal{F}$-distance $\mathcal{O}(\Delta)$, where $\Delta = \Delta_{\min} \cdot \sqrt{\log(1/\beta)}$ is only a small factor above the lower bound. The number of batches it requires for any VC family $\mathcal{F}$ is only a logarithmic factor more than needed to achieve the same error without adversarial batches, showing that robustness can be incorporated at little extra cost. This provides the first demonstration that distributions can be learned (1) robustly and (2) sample-efficiently, over infinite, and even continuous domains.

As expected from the setting's vast generality, as in the non-adversarial setting, for some VC families, one cannot expect to find a computationally efficient algorithm. We, therefore, consider a natural and important VC family over the reals that, as we shall soon see, translates into efficient and robust algorithms for TV-learning and classification over $\mathbb{R}$.

Let $\mathcal{F}_k$ be the family of all unions of at most $k$ intervals over $\mathbb{R}$. We derive a computationally efficient algorithm that estimates distributions to $\mathcal{F}_k$-distance $\mathcal{O}(\Delta)$ using only $\tilde{\mathcal{O}}(1/\Delta)$ times more samples than the non-adversarial, or information-theoretic adversarial cases.

Building on these techniques, we return to estimation in total variation (TV) distance. We consider the family of distributions whose Yatracos Class [Yat85] have finite VC dimension. This family consists of both discrete and continuous distributions, and includes piecewise polynomials, Gaussians in one or more dimensions, and arguably most practical distribution families. We show that all these distributions can be learned robustly from batches to a TV distance $\mathcal{O}(\Delta)$, which is only a factor $\sqrt{\log(1/\beta)}$ above the adversarial TV-distance lower bound of $\Delta_{\min}$. It also achieves sample complexity that is at most a logarithmic factor more than required for non-adversarial case.

These results too are very general, hence as in the non-adversarial case, one cannot expect a computationally efficient algorithm for all cases. We therefore consider the natural and important general class $\mathcal{P}_{t,d}$ of $t$-piecewise degree-$d$ polynomial distributions over the reals.

To agnostically learn distributions $\mathcal{P}_{t,d}$, we combine the results above with an existing, non-adversarial, polynomial-learning algorithm [ADLS17]. We derive a polynomial-time algorithm for estimating polynomials in $\mathcal{P}_{t,d}$ to a TV distance $\mathcal{O}(\Delta)$. The algorithm's sample complexity is linear in $td$, which is the best possible, and similar to learning in $\mathcal{F}_k$-distance, only $\tilde{\mathcal{O}}(1/\Delta)$ times above the non-adversarial, or information-theoretic adversarial sample complexity.

This is the first algorithm that achieves polynomial sample and time complexity for robust learning for this class, and the first that applies to the non-finite domains.

The general formulation also allows us to use batch-structure for robustness in other learning tasks. We apply this framework to derive the first robust agnostic classifiers. The goal is to minimize the excess loss in comparison to the best hypothesis, in the presence of adversarial batches.

We first modify the lower bound on distribution learning to show that any classification algorithm with adversarial batches must incur an excess loss $\mathcal{O}(\Delta_{\min})$, even with the number of batches $\to \infty$. We then derive a general algorithm that achieves additive excess loss $\mathcal{O}(\Delta)$ for general binary classification using a number of samples that is again only a logarithmic factor larger than required to achieve the same excess loss in the non-adversarial setting.

Finally, we consider classification over $\mathbb{R}$. Many natural and practical classifiers have decision regions consisting of finitely many disjoint intervals. We apply the above results to derive a computationally efficient algorithm for hypotheses consisting of $k$ intervals. Similar to previous results, its sample complexity is linear in $k$ and only a factor $\mathcal{O}(1/\Delta)$ larger than required in the non-adversarial setup.

The rest of the paper is organized as follows. Section 2 describes the main technical results and their applications to distribution estimation and classification. Section 3 discusses the other related work. Section 4 provides an overview of the filtering algorithm that enables these results. Proofs and more details are relegated to the appendix.

## 2  Results

We consider learning from batches of samples, when a $\beta-$fraction of batches are adversarial.

More precisely, $B$ is a collection of $m$ batches, composed of two *unknown* sub-collections. A *good sub-collection* $B_G \subseteq B$ of $\geq (1-\beta)m$ *good batches*, where each batch $b$ consists of $n$ independent samples from a common distribution $p$ over $\Omega$. And an *adversarial sub-collection* $B_A = B \setminus B_G$ of the remaining $\leq \beta m$ batches, each consisting of the same number $n$ of arbitrary $\Omega$ elements, that for simplicity we call *samples* as well. Note that the adversarial samples may be chosen in any way, including after observing the good samples.

The next subsection describes the main technical results for learning in $\mathcal{F}$ distance. Subsequent subsections apply these results to learn distributions in TV distance and to achieve robust binary classification.

### 2.1  Estimating distributions in $\mathcal{F}$ distance

Our goal is to use samples generated by a target distribution $p$ to approximate it to a small $\mathcal{F}$-distance. For general families $\mathcal{F}$, this goal cannot be accomplished even with just good batches. Let $\mathcal{F} = \Sigma$ be the collection of all subsets of the real interval domain $\Omega = [0,1]$. For any total number $t$ of samples, with high probability, it is impossible to distinguish the uniform distribution over $[0,1]$ from a uniform discrete distribution over a random collection of $\gg t^2$ elements in $[0,1]$. Hence any estimator must incur TV-distance 1 for some distribution.

This difficulty is addressed by Vapnik-Chervonenkis (VC) Theory. The collection $\mathcal{F}$ *shatters* a subset $S \subseteq \Omega$ if every subset of $S$ is the intersection of $S$ with a subset in $\mathcal{F}$. The VC-dimension $V_{\mathcal{F}}$ of $\mathcal{F}$ is the size of the largest subset shattered by $\mathcal{F}$.

Let $X^t = X_1, \dots, X_t$, be i.i.d. samples from a distribution $p$. The empirical probability of $S \subseteq \Omega$ is

$$\bar{p}_t(S) := |\{i : X_i \in S\}|/t.$$

The fundamental *Uniform deviation inequality* of VC theory [VC71, Tal94] states that if $\mathcal{F}$ has finite VC-dimension $V_{\mathcal{F}}$, then $\bar{p}_t$ estimates $p$ well in $\mathcal{F}$ distance. For all $\delta > 0$, with probability $> 1 - \delta$,

$$||p - \bar{p}_t||_{\mathcal{F}} \leq \mathcal{O}\big(\sqrt{(V_{\mathcal{F}} + \log 1/\delta)/t}\,\big).$$

The above is also the lowest achievable $\mathcal{F}$-distance, hence we call it the *information-theoretic limit*.

In the adversarial-batch scenario, a fraction $\beta$ of the batches may be corrupted. It is easy to see that for any number $m$ of batches, however large, the adversary can cause $\bar{p}_t$ to approximate $p$ to $\mathcal{F}$-distance $\geq \beta/2$, namely $||\bar{p}_t - p||_{\mathcal{F}} \geq \beta/2$.

Let $\bar{p}_{B'}$ be the empirical distribution induced by the samples in a collection $B' \subseteq B$. Our first result states that if $\mathcal{F}$ has a finite VC-dimension, for total samples $m \cdot n \geq \tilde{\mathcal{O}}(V_{\mathcal{F}}/\Delta^2)$, the batches in $B$ can be "cleaned" to a sub-collection $B^*$ where $||p - \bar{p}_{B^*}||_{\mathcal{F}} = \mathcal{O}(\Delta)$, namely, a simple empirical estimator of the samples in $B^*$ recovers $p$ to a small $\mathcal{F}$-distance.

**Theorem 1.** *For any $\mathcal{F}$, $\beta \leq 0.4$, $\delta > 0$, and $mn \geq \tilde{\mathcal{O}}\left(\frac{V_{\mathcal{F}} + \log 1/\delta}{\Delta^2}\right)$, there is an algorithm that w.p. $\geq 1-\delta$ returns a sub-collection $B^* \subseteq B$ s.t. $|B^* \cap B_G| \geq (1 - \frac{\beta}{6})|B_G|$ and $||p - \bar{p}_{B^*}||_{\mathcal{F}} \leq \mathcal{O}(\Delta)$.*

The $\mathcal{F}$-distance bound matches the lower bound $\Delta_{\min}$ up to a small $\mathcal{O}(\sqrt{\log(1/\beta)})$ factor. The number $m \cdot n$ of samples required to achieve this estimation error are the same (up to a logarithmic factor) as the minimum required to achieve the same estimation error even for the non-adversarial setting.

The theorem applies to all families with finite VC dimension, and like most other results of this generality, it is necessarily non-constructive in nature. Yet it provides a road map for constructing efficient algorithms for many specific natural problems. In Section 4 we use this approach to derive a polynomial-time algorithm that learns distributions with respect to one of the most important and practical VC classes, where $\Omega = \mathbb{R}$, and $\mathcal{F} = \mathcal{F}_k$ is the collection of all unions of at most $k$ intervals.

**Theorem 2.** *For any $k > 0$, $\beta \leq 0.4$, $\delta > 0$, and $mn \geq \tilde{\mathcal{O}}\left(\frac{k + \log 1/\delta}{\Delta^3}\right)$, there is an algorithm that runs in time polynomial in all parameters, and with probability $\geq 1 - \delta$ returns a sub-collection $B^* \subseteq B$, such that $|B^* \cap B_G| \geq (1 - \frac{\beta}{6})|B_G|$ and $||p - \bar{p}_{B^*}||_{\mathcal{F}_k} \leq \mathcal{O}(\Delta)$.*

The above polynomial-time algorithm can achieve $\mathcal{F}_k$ error $\Delta$ using the number of samples only $\tilde{\mathcal{O}}(1/\Delta)$ times the minimum required to achieve the same estimation error by any algorithm even for the non-adversarial setting. Note that the sample complexity in both Theorems 1 and 2 are independent of the domain size and depends linearly on the VC dimension of the subset family.

Section 4 provides a short overview of the algorithms used in the above theorems. The complete algorithms and proof of the two theorems appear in the appendix.

## 2.2 Learning distributions in total-variation distance

Our ultimate objective is to estimate the target distribution in total variation (TV) distance, one of the most common measures in distribution estimation. In this and the next subsection, we follow a framework developed in [DL01], see also [Dia16].

As noted earlier, the sample complexity of estimating distributions in TV-distance grows with the domain size, becoming infeasible for large discrete domains and impossible for continuous domains. A natural approach to address this intractability is to assume that the underlying distribution belongs to, or is near, a structured class $\mathcal{P}$ of distributions.

Let $\mathrm{opt}_{\mathcal{P}}(p) := \inf_{q \in \mathcal{P}} ||p - q||_{TV}$ be the TV-distance of $p$ from the closest distribution in $\mathcal{P}$. For example, for $p \in \mathcal{P}$, $\mathrm{opt}_{\mathcal{P}}(p) = 0$. Given $\epsilon, \delta > 0$, we try to use samples from $p$ to find an estimate $\hat{p}$ such that, with probability $\geq 1 - \delta$,

$$||p - \hat{p}||_{TV} \leq \alpha \cdot \mathrm{opt}_{\mathcal{P}}(p) + \epsilon$$

for a universal constant $\alpha \geq 1$, namely, to approximate $p$ about as well as the closest distribution in $\mathcal{P}$.

Following [DL01], we utilize a connection between distribution estimation and VC dimension. Let $\mathcal{P}$ be a class of distributions over $\Omega$. The *Yatracos class* [Yat85] of $\mathcal{P}$ is the family of $\Omega$ subsets

$$\mathcal{Y}(\mathcal{P}) := \{\{\omega \in \Omega : p(\omega) \geq q(\omega)\} : p, q \in \mathcal{P}\}.$$

It is easy to verify that for distributions $p, q \in \mathcal{P}$, $||p - q||_{TV} = ||p - q||_{\mathcal{Y}(\mathcal{P})}$. The *Yatracos minimizer* of a distribution $p$ is its closest distribution, by $\mathcal{Y}(\mathcal{P})$-distance, in $\mathcal{P}$,

$$\psi_{\mathcal{P}}(p) := \arg\min_{q \in \mathcal{P}} ||q - p||_{\mathcal{Y}(\mathcal{P})},$$

where ties are broken arbitrarily. Theorem 6.3 in [DL01] uses these definitions and a sequence of triangle inequalities to show that for any distributions $p, p'$, and any distribution class $\mathcal{P}$,

$$||p - \psi_{\mathcal{P}}(p')||_{TV} \leq 3 \cdot \mathrm{opt}_{\mathcal{P}}(p) + 4||p - p'||_{\mathcal{Y}(\mathcal{P})}. \tag{1}$$

Therefore, given a distribution $p'$ that approximates $p$ in $\mathcal{Y}(\mathcal{P})$-distance, its Yatracos minimizer $\psi_{\mathcal{P}}(p')$ approximates $p$ in TV-distance.

If the Yatracos class $\mathcal{Y}(\mathcal{P})$ has finite VC dimension, the VC-bound ensures that for the empirical distribution $\bar{p}_t$ of $t$ i.i.d. samples from $p$, $||\bar{p}_t - p||_{\mathcal{Y}(\mathcal{P})}$ decreases to zero as $t$ increases, and $\psi_{\mathcal{P}}(\bar{p}_t)$ can be used to approximate $p$ in TV-distance. This general method has led to many sample-and computationally-efficient algorithms for estimating structured distributions, *e.g.,* [ADLS17].

However, as discussed earlier, with a $\beta$-fraction of adversarial batches, the empirical distribution of all samples can be at a $\mathcal{Y}(\mathcal{P})$-distance as large as $\Theta(\beta)$ from $p$, leading to a large TV-distance.

Yet Theorem 1 shows that data can be "cleaned" to remove outlier batches and retain batches $B^* \subseteq B$ whose empirical distribution $\bar{p}_{B^*}$ approximates $p$ to a much smaller $\mathcal{Y}(\mathcal{P})$-distance of $\mathcal{O}(\Delta)$. Letting $p^* = \psi_{\mathcal{P}}(\bar{p}_{B^*})$ and using Equation (1), we obtain a much better approximation of $p$ in TV distance.

**Theorem 3.** *For a distribution class $\mathcal{P}$ with Yatracos Class of finite VC dimension $v$, for any $\beta \leq 0.4$, $\delta > 0$, and $mn \geq \tilde{\mathcal{O}}\left(\frac{v + \log 1/\delta}{\Delta^2}\right)$, there is an algorithm that w. p. $\geq 1 - \delta$ returns a distribution $p^* \in \mathcal{P}$ such that $||p - p^*||_{TV} \leq 3 \cdot opt_{\mathcal{P}}(p) + \mathcal{O}(\Delta)$.*

The estimation error achieved in the theorem for TV-distance matches the lower bound to a small log factor of $O(\sqrt{\log(1/\beta)})$, and is valid for any class $\mathcal{P}$ with finite VC Dimensional Yatracos Class.

Moreover, the upper bound on the number of samples (or batches) required by the algorithm to estimate $p$ to the above distance matches a similar general upper bound obtained for non-adversarial setting to a log factor. This results for the first time shows that it is possible to learn a wide variety of distributions robustly using batches, even over continuous domains.

## 2.3 Learning univariate structured distributions

We apply the general results in the last two subsections to estimate distributions over the real line. We focus on one of the most studied, and important, distribution families, the class of piecewise-polynomial distributions. A distribution $p$ over $[a, b]$ is $t$-piecewise, degree-$d$, if there is a partition of $[a, b]$ into $t$ intervals $I_1, \ldots, I_t$, and degree-$d$ polynomials $r_1, \ldots, r_t$ such that $\forall j$ and $x \in I_j$, $p(x) = r_j(x)$. The definition extends naturally to finite distributions over $[k] = \{1, \ldots, k\}$.

Let $\mathcal{P}_{t,d}$ denote the collection of all $t$-piecewise degree $d$ distributions. $\mathcal{P}_{t,d}$ is interesting in its own right, as it contains important distribution classes such as histograms. In addition, it approximates other important distribution classes, such as monotone, log-concave, Gaussians, and their mixtures, arbitrarily well, *e.g.,* [ADLS17].

Note that for any two distributions $p, q \in \mathcal{P}_{t,d}$, the difference $p - q$ is a $2t$-piecewise degree-$d$ polynomial, hence every set in the Yatracos class of $\mathcal{P}_{t,d}$, is the union of at most $2t \cdot d$ intervals in $\mathbb{R}$. Therefore, $\mathcal{Y}(\mathcal{P}_{t,d}) \subseteq \mathcal{F}_{2t \cdot d}$. And since $V_{\mathcal{F}_k} = O(k)$ for any $k$, $\mathcal{Y}(\mathcal{P}_{t,d})$ has VC dimension $\mathcal{O}(td)$.

Theorem 3 can then be applied to show that any target distribution $p$ can be estimated by a distribution in $\mathcal{P}_{t,d}$ to a TV-distance $\Delta$, using a number of samples, that is within a logarithmic factor from the minimum required [CDSS14] even when all samples are i.i.d. generated from $p$.

**Corollary 4.** *For any distribution $p$ over $\mathbb{R}$, $t$, $d$, $\beta \leq 0.4$, $\delta > 0$, and $mn \geq \tilde{\mathcal{O}}\left(\frac{td + \log 1/\delta}{\Delta^2}\right)$, there is an algorithm that with probability $\geq 1 - \delta$ returns a distribution $p^* \in \mathcal{P}_{t,d}$ such that $||p - p^*||_{TV} \leq 3 \cdot opt_{\mathcal{P}_{t,d}}(p) + \mathcal{O}(\Delta)$.*

Next we provide a polynomial-time algorithm for estimating $p$ to the same $\mathcal{O}(\Delta)$ TV-distance, but with an extra $\tilde{\mathcal{O}}(1/\Delta)$ factor in sample complexity.
Theorem 2 provides a polynomial time algorithm that returns a sub-collection $B^* \subseteq B$ of batches whose empirical distribution $\bar{p}_{B^*}$ is close to $p$ in $\mathcal{F}_{2td}$-distance. [ADLS17] provides a polynomial time algorithm that for any distribution $q$ returns a distribution in $p' \in \mathcal{P}_{t,d}$ minimizing $||p' - q||_{\mathcal{F}_{2td}}$ to a small additive error. Then equation (1) and Theorem 2 yield the following result. We provide formal proof of the theorem in the appendix.

**Theorem 5.** *For any distribution $p$ over $\mathbb{R}$, $n$, $m$, $\beta \leq 0.4$, $t$, $d$, $\delta > 0$, and $mn \geq \tilde{\mathcal{O}}\left(\frac{td + \log 1/\delta}{\Delta^3}\right)$, there is a polynomial time algorithm that w. p. $\geq 1 - \delta$ returns a distribution $p^* \in \mathcal{P}_{t,d}$ such that $||p - p^*||_{TV} \leq \mathcal{O}(opt_{\mathcal{P}_{t,s}}(p)) + \mathcal{O}(\Delta)$.*

## 2.4 Binary classification

Our framework extends beyond distribution estimation. Here we describe its application to Binary classification. Consider a family $\mathcal{H} : \Omega \rightarrow \{0, 1\}$ of Boolean functions, and a distribution $p$ over $\Omega \times \{0, 1\}$. Let $(X, Y) \sim p$, where $X \in \Omega$ and $Y \in \{0, 1\}$. The loss of classifier $h \in \mathcal{H}$ for distribution $p$ is $r_p(h) := \Pr_{(X,Y) \sim p}[h(X) \neq Y]$. The *optimal classifier* for distribution $p$ is $h^{\text{opt}}(p) := \arg\min_{h \in \mathcal{H}} r_p(h)$, and hence the *optimal loss* is $r_p^{\text{opt}}(\mathcal{H}) := r_p(h^{\text{opt}}(p))$. The goal is to return a classifier $h \in \mathcal{H}$ whose *excess loss* $r_p(h) - r_p^{\text{opt}}(\mathcal{H})$ compared to the optimal loss is small.

Consider the following natural extension of VC-dimension from families of subsets to families of Boolean functions. For a boolean-function family $\mathcal{H}$, define the family
$$\mathcal{F}_{\mathcal{H}} := \{(\{\omega \in \Omega : h(\omega) = y\}, \bar{y}) : h \in \mathcal{H}, y \in \{0, 1\}\}$$
of subsets of $\Omega \times \{0, 1\}$, and let the VC dimension of $\mathcal{H}$ be $V_{\mathcal{H}} := V_{\mathcal{F}_{\mathcal{H}}}$.

The next simple lemma, proved in the appendix, upper bounds the excess loss of the optimal classifier in $\mathcal{H}$ for a distribution $q$ for another distribution $p$ in terms of $\mathcal{F}_{\mathcal{H}}$ distance between the distributions.
**Lemma 6.** *For any class $\mathcal{H}$ and distributions $p$ and $q$, $r_p(h^{opt}(q)) - r_p^{opt}(\mathcal{H}) \leq 4||p - q||_{\mathcal{F}_{\mathcal{H}}}$.*

When $q$ is an empirical distribution of the samples, $h^{\text{opt}}(q)$ is called the *empirical-risk minimizer*. If $q$ is the empirical distribution of the samples generated i.i.d. from $p$, from VC inequality, the excess loss of the empirical-risk minimizer in the above equation goes to zero if VC dimension of $\mathcal{H}$ is finite.

Yet as discussed earlier, when a $\beta$-fractions of the batches, and hence samples, are chosen by an adversary, the empirical distribution of all samples can be at a large $\mathcal{F}_{\mathcal{H}}$-distance $\mathcal{O}(\beta)$ from $p$, leading to an excess-classification-loss up to $\Omega(\beta)$ for the empirical-risk minimizer.

Theorem 1 states that the collection of batches can be "cleaned" to obtain a sub-collection $B^* \subseteq B$ whose empirical distribution has a lower $\mathcal{F}_{\mathcal{H}}$-distance from $p$. The above lemma then implies that the optimal classifier $h^{\text{opt}}(\bar{p}_{B^*})$ for the empirical distribution $\bar{p}_{B^*}$ of the cleaner batches will have a small-excess-classification-loss for $p$ as well. The resulting non-constructive algorithm has excess-classification-loss and sample complexity that are optimal to a logarithmic factor.

**Theorem 7.** *For any $\mathcal{H}$, $p$, $\beta \leq 0.4$, $\delta > 0$, and $mn \geq \tilde{\mathcal{O}}\left(\frac{V_{\mathcal{H}} + \log 1/\delta}{\Delta^2}\right)$, there is an algorithm that with probability $\geq 1 - \delta$ returns a classifier $h^*$, whose excess lose is $r_p(h^*) - r_p^{opt}(\mathcal{H}) \leq \mathcal{O}(\Delta)$.*

To complement this result, we show an information-theoretic lower bound of $\Omega(\Delta_{\min})$ on the excess loss. The proof is in the appendix. Recall that a similar lower bound holds for learning distribution.

**Theorem 8.** *For any $\beta$, $n$, and $\mathcal{H}$ s.t. $V_{\mathcal{H}} \geq 1$, there are a distribution $p$ and an adversary, such that any algorithm, with probability $\geq 1/2$, incurs an excess loss $\Omega(\Delta_{min})$, even as number of batches $m \to \infty$.*

To derive a computationally-efficient algorithm, we focus on the following class of binary functions. For $k \geq 1$, let $\mathcal{H}_k$ denote the collection of all binary functions over $\mathbb{R}$ whose decision region, namely values mapping to 0 or 1, consists of at most $k$-intervals. The VC dimension of $\mathcal{F}_{\mathcal{H}_k}$ is clearly $\mathcal{O}(k)$.

Theorem 2 describes a polynomial-time algorithm that returns a cleaner data w.r.t. $\mathcal{F}_{\mathcal{H}_k}$ distance. From Lemma 6, the classifier that minimizes the loss for the empirical distribution of this cleaner data will have a small excess loss. Furthermore, [Maa94] derived a polynomial-time algorithm to find the empirical risk minimizer $h \in \mathcal{H}_k$ for any given samples. Combining these results, gives a robust computationally efficient classifier in $\mathcal{H}_k$. We provide a formal proof in the appendix.

**Theorem 9.** *For any $k$, $p$, $\beta \leq 0.4$, $\delta > 0$, and $mn \geq \tilde{\mathcal{O}}\left(\frac{k + \log 1/\delta}{\Delta^3}\right)$, there is a polynomial-time algorithm that w. p. $\geq 1 - \delta$ returns a classifier $h^*$, whose excess loss is $r_p(h^*) - r_p^{opt}(\mathcal{H}_k) \leq \mathcal{O}(\Delta)$.*

## 3 Other related and concurrent work

The current results extend several long lines of work on estimating structured distributions, including [O'B16, Dia16, AM18, ADLS17]. The results also relate to classical robust-statistics work [Tuk60, Hub92]. There has also been significant recent work leading to practical distribution learning algorithms that are robust to adversarial contamination of the data. For example, [DKK+16, LRV16] presented algorithms for learning the mean and covariance matrix of high-dimensional sub-gaussian and other distributions with bounded fourth moments in presence of the adversarial samples. Their estimation guarantees are typically in terms of $L_2$, and do not yield the $L_1$- distance results required for discrete distributions. The work was extended in [CSV17] to the case when more than half of the samples are adversarial. Their algorithm returns a small set of candidate distributions one of which is a good approximate of the underlying distribution. The filtering based method has also played a key role in other robust learning algorithms in high dimension [DKK+17, DKK+18, SCV17, DKK+19]. These works apply filtering on samples instead on batches of samples, as in [JO19] and in this paper, and recover in a different metric. For a more extensive survey on robust learning algorithms see [SCV17, DKK+19].

Another motivation for this work derives from the practical federated-learning problem, where information arrives in batches [MMR+16, MR17].

**Concurrent work** Concurrent to our work, [CLM20] also extends the filtering algorithm of [JO19] to obtain robust batch learning algorithms for estimating piecewise polynomials. They derive a polynomial-time algorithm that learns distributions in $\mathcal{P}_{t,d}$ over a finite domain $[k]$ to the same TV distance $\mathcal{O}(\Delta)$ as we do, but requires $\tilde{\mathcal{O}}((td)^2 \log^3(k)/\Delta^2)$ samples, where $\tilde{\mathcal{O}}$ hides a logarithmic factor in $1/\Delta$. In contrast, our results show that this accuracy can be achieved using $\tilde{\mathcal{O}}(td/\Delta^2)$ samples, and by a polynomial-time algorithm with sample complexity is $\tilde{\mathcal{O}}(td/\Delta^3)$. Importantly, our algorithms' complexity does not depend on the alphabet size $[k]$, which allows us to extend them to more general non-finite and even continuous domains. In addition, we considered other distribution classes and learning tasks such as classification.

Another concurrent work [KFAL20] focuses on the sample complexity of robust batch classification using adversarial batches. Their results achieve an excess loss of $\mathcal{O}(\sqrt{V_{\mathcal{H}}} \cdot \Delta)$, where $V_{\mathcal{H}}$ is the VC-dimension of the hypothesis class, whereas we achieve an excess loss only $\mathcal{O}(\Delta)$.

# 4 Overview of the filtering framework for learning in $\mathcal{F}$ distance

To derive both the information-theoretic and computationally-efficient algorithms for general robust learning from batches, we generalize a finite filtering-based approach in [JO19]. We first describe the original algorithm and outline how it can be extended to general learning problems. A more complete and formal presentation appears in the appendix.

Recall that $B$ is the collection of all $m$ batches and each batch $b \in B$ has $n$ samples from the domain $\Omega$. A batch $b$ estimates the probability $p(S)$ of a subset $S \in \Sigma$ by its empirical probability. Each subset $S \in \Sigma$, assigns to every batch $b \in B$, a *corruption score* $\psi_b(S)$, defined in the appendix, based on how far the batch's estimate of $p(S)$ is from the median of the estimates for all batches. Similarly, each subset $S$ assigns to every sub-collection $B' \subseteq B$ of batches a corruption score $\psi_{B'}(S) := \sum_{b \in B'} \psi_b(S)$, the sum of individual corruption score of each batch.

We first describe a general *filtering* approach to robust learning from batches. A collection $C \subseteq \Sigma$ of subsets, is *learnable via filtering* if one can "filter out" bad batches in $B$ and find a "good" subset $B^* \subseteq B$ of batches that approximates $p$ to a small $C$-distance,

$$||p - \bar{p}_{B^*}||_C = \max_{S \in C} |p(S) - \bar{p}_{B^*}(S)| \leq \mathcal{O}(\Delta). \tag{2}$$

We describe two properties ensuring that $C$ is learnable via filtering. A finite $C \subseteq \Sigma$ is learnable via filtering if there is a threshold $\tau$ such that all subsets $S \in C$ and all sub-collection $B' \subseteq B$ that contain most good batches, namely $|B' \cap B_G| \geq (1 - \beta/6)|B_G|$, satisfy the following two properties:

1. If the corruption score is low, $\psi_{B'}(S) < \tau$, then $B'$ estimates $p(S)$ well, $|p(S) - \bar{p}_{B'}(S)| = \mathcal{O}(\Delta)$.
2. If $\psi_{B'}(S) > \tau$, then there is a (probabilistic) method that removes batches in $B'$, while ensuring that and each batch removed is adversarial with probability at least 0.95, until $\psi_{B'}(S) < \tau$.

A simple algorithm shows that these two properties imply that $C$ is learnable by filtering. Start with $B' = B$, find a filter $S \in C$ with $\psi_{B'}(S) > \tau$, remove the batches from $B'$, and repeat the process until the corruption is small, $\psi_{B'}(S) < \tau$, for all filters in $C$. By property 2, each deleted batch is adversarial with probability $> 0.95$. Since there are at most $\beta m$ adversarial batches, w.h.p. at most $0.1\beta m$ good batches are deleted. Consequently $|B' \cap B_G| \geq (1 - \beta/6)|B_G|$. By property 1, when the algorithm ends, $B^* = B'$ achieves (2).

While this algorithm describes the core of the technique, three significant challenges remain.

The above algorithm applies for finite classes $C$. However, the VC class $\mathcal{F}$ may be infinite, or even uncountable. To apply the algorithm we need to find a finite subset $C$ such that learning in $C$ distance implies learning in $\mathcal{F}$ distance. In the appendix, we prove an essential *Robust Covering Theorem*, showing that for an appropriate $\epsilon$, letting $C$ be an $\epsilon$-cover of $\mathcal{F}$ under empirical density $\bar{p}_B$, suffices to learn $p$ in $\mathcal{F}$ distance. This is despite the fact that a fraction $\beta$ of the batches in $B$ may be adversarially chosen, and even depend on good samples.

The next key challenge is to show that the two properties hold for all subsets in the $\epsilon$-cover. We establish this fact by showing that with sufficiently many batches, w.h.p., the two properties hold for all subsets $S \in \mathcal{F}$. The proof requires addressing additional technical challenges, as number of subsets in $\mathcal{F}$ could be infinite.

Choosing any finite $\epsilon$-cover $C \subseteq \mathcal{F}$ under density $\bar{p}_B$, therefore yields an information-theoretic algorithm with near-optimal sample complexity. This gives us the near sample optimal algorithm in Theorem 1. However, computationally-efficient algorithms pose one additional challenge. The size of $C$ may be exponential in the VC dimension, and hence searching for a subset in $C$ with a high corruption score may be computationally infeasible.

For the VC class $\mathcal{F}_k$, we overcome this difficulty by choosing the set $C$ of filters from a larger class than $\mathcal{F}_k$ itself so that that still obeys the two properties, but allows for an efficient search. Though $C$ is chosen from a larger class, we ensure that the sample complexity increase is small. Specifically, we let $C$ be the collection of all subsets of a $k'$-partition of $\Omega$, for an appropriate $k'$ that is linear in $k$. Subsets in such a cover $C$ correspond to binary vectors in $\{0,1\}^{k'}$. A novel semi-definite-programming based algorithm derived in [JO19] finds a subset $S \in C$ with nearly the highest corruption $\psi_{B'}(S)$ in time only polynomial in $k'$. This allows us to obtain the polynomial-time algorithm in Theorem 2.

To summarize, this universal filtering approach allows us to "clean" the data and enables the general robust distribution estimators and classifiers we construct.

*Remark.* In some applications the distributions underlying genuine batches may differ from the common target distribution $p$ by a small TV distance, say $\eta > 0$. For simplicity, in this paper we presented the analysis for $\eta = 0$, where all the good batches have the same distribution $p$. For $\eta > 0$, even for binary alphabets, [QV17] derived the adversarial batch lower bound of $\Omega(\eta + \beta/\sqrt{n})$ on TV distance. And even the trivial empirical estimator achieves $\mathcal{O}(\eta + \beta)$ TV-error, which has optimal linear dependence on $\eta$. Therefore, filtering algorithms do not need to do anything sophisticated for general $\eta$ and incurs only an extra $\mathcal{O}(\eta)$ error as noted in [JO19] for unstructured distributions, and the same holds for our algorithms for learning structured distributions and binary classification.

## Broader impact

With the vast increase in data availability, data sources are often corrupt or untrustworthy. This untrusted data severely limits the efficacy of several leaning algorithms, even when a vast amount of the data is available. Yet in many applications the data is collected in batches. We consider two essential problems in machine learning, Classification and distribution estimation. We show that in these applications, the effect of data corruption diminishes with the batch size, and demonstrate how batch structure can be used to reduce the effect of the corrupted or adversarial data, thereby paving a path to more reliable machine learning algorithms.

## Acknowledgements

We thank Vaishakh Ravindrakumar and Yi Hao for helpful comments in the prepration of this manuscript and the authors of the concurrent work [CLM20] for coordinating submission with us.

We are grateful to the National Science Foundation (NSF) for supporting this work through grants CIF-1564355 and CIF-1619448.

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
