[Supplementary Material]

The Appendix is organized as follows: Section A introduces notation and states some useful facts. Section B recounts basic tools from VC theory used to derive the results. Section C derives a framework for robust distribution estimation in $\mathcal{F}$-distance and proves Theorem 1. Building on this framework it then develops computationally efficient algorithms for learning in $\mathcal{F}_k$ distance and proves Theorem 2. Section D gives the proof of the filtration properties and other results used in Section C. Section E gives the other remaining proofs of the main paper.

# A    Preliminaries

We introduce terminology that helps describe the approach and results. Some of the work builds on results in [JO19], and we keep the notation consistent.

Recall that $B$, $B_G$, and $B_A$ are the collections of all-, good-, and adversarial-batches. Let $B' \subseteq B$, $B'_G \subseteq B_G$, and $B'_A \subseteq B_A$, denote sub-collections of all-, good-, and bad-batches. We also let $S$ denote a subset in the Borel $\sigma$-field $\Sigma$ on domain $\Omega$.

Let $X^b_1, X^b_2, ..., X^b_n$ denote the $n$ samples in a batch $b$, and let $\mathbf{1}_S$ denote the indicator random variable for a subset $S \in \Sigma$. Every batch $b \in B$ induces an empirical measure $\bar{\mu}_b$ over the domain $\Omega$, where for each $S \in \Sigma$,

$$\bar{\mu}_b(S) := \frac{1}{n} \sum_{i \in [n]} \mathbf{1}_S(X^b_i).$$

Similarly, any sub-collection $B' \subseteq B$ of batches induces an empirical measure $\bar{p}_{B'}$ defined by

$$\bar{p}_{B'}(S) := \frac{1}{|B'|n} \sum_{b \in B'} \sum_{i \in [n]} \mathbf{1}_S(X^b_i) = \frac{1}{|B'|} \sum_{b \in B'} \bar{\mu}_b(S).$$

We use two different symbols to denote empirical distribution defined by single batch and a sub-collection of batches to make them easily distinguishable. Note that $\bar{p}_{B'}$ is the mean of the empirical measures $\bar{\mu}_b$ defined by the batches $b \in B'$.

Recall that $n$ is the batch size. For $r \in [0, 1]$, let $\mathrm{V}(r) := \frac{r(1-r)}{n}$, the variance of a Binomial$(r, n)$ random variable. Observe that

$$\forall\, r, s \in [0, 1],\ \mathrm{V}(r) \leq \frac{1}{4n} \quad \text{and} \quad |\mathrm{V}(r) - \mathrm{V}(s)| \leq \frac{|r - s|}{n}, \tag{3}$$

where the second property follows as $|r(1 - r) - s(1 - s)| = |r - s| \cdot |1 - (r + s)| \leq |r - s|$.

For $b \in B_G$, the random variables $\mathbf{1}_S(X^b_i)$ for $i \in [n]$ are distributed i.i.d. Bernoulli$(p(S))$, and since $\bar{\mu}_b(S)$ is their average,

$$E[\,\bar{\mu}_b(S)\,] = p(S) \quad \text{and} \quad \mathrm{Var}[\,\bar{\mu}_b(S)\,] = E[(\bar{\mu}_b(S) - p(S))^2] = \mathrm{V}(p(S)).$$

For batch collection $B' \subseteq B$ and subset $S \in \Sigma$, the empirical probability $\bar{\mu}_b(S)$ of $S$ will vary with the batch $b \in B'$. The *empirical variance* of these empirical probabilities is

$$\overline{\mathrm{V}}_{B'}(S) := \frac{1}{|B'|} \sum_{b \in B'} (\bar{\mu}_b(S) - \bar{p}_{B'}(S))^2.$$

# B    Vapnik-Chervonenkis (VC) theory

We recall some basic concepts and results in VC theory, and derive some of their simple consequences that we use later in deriving our main results.

The *VC shatter coefficient* of $\mathcal{F}$ is

$$S_{\mathcal{F}}(t) := \sup_{x_1, x_2, .., x_t \in \Omega} |\{\{x_1, x_2, .., x_t\} \cap S : S \in \mathcal{F}\}|,$$

the largest number of subsets of $t$ elements in $\Omega$ obtained by intersections with subsets in $\mathcal{F}$. The VC dimension of $\mathcal{F}$ is

$$V_{\mathcal{F}} := \sup\{t : S_{\mathcal{F}}(t) = 2^t\},$$

the largest number of $\Omega$ elements that are "fully shattered" by $\mathcal{F}$. The following Lemma [DL01] bounds the Shatter coefficient for a VC family of subsets.

**Lemma 10** ([DL01]). *For all $t \geq V_{\mathcal{F}}$, $\quad S_{\mathcal{F}}(t) \leq \left(\frac{t\,e}{V_{\mathcal{F}}}\right)^{V_{\mathcal{F}}}$.*

Next we state the VC-inequality for relative deviation [VC74, AST93].

**Theorem 11.** *Let $p$ be a distribution over $(\Omega, \Sigma)$, and $\mathcal{F}$ be a VC-family of subsets of $\Omega$ and $\bar{p}_t$ denote the empirical distribution from $t$ i.i.d samples from $p$. Then for any $\epsilon > 0$, with probability $\geq 1 - 8S_{\mathcal{F}}(2t)e^{-t\epsilon^2/4}$,*

$$\sup_{S \in \mathcal{F}} \max\left\{ \frac{\bar{p}_t(S) - p(S)}{\sqrt{\bar{p}_t(S)}}, \frac{p(S) - \bar{p}_t(S)}{\sqrt{p(S)}} \right\} \leq \epsilon.$$

Another important ingredient commonly used in VC Theory is the concept of covering number that reflects the smallest number of subsets that approximate each subset in the collection.

Let $p$ be any probability measure over $(\Omega, \Sigma)$ and let $\mathcal{F} \subseteq \Sigma$ be a family of subsets. A collection $\mathcal{C} \subseteq \Sigma$ of subsets is an $\epsilon$-*cover* of $\mathcal{F}$ under distribution $p$ if for any $S \in \mathcal{F}$, there exists a $S' \in \mathcal{C}$ with $p(S \triangle S') \leq \epsilon$. The $\epsilon$-*covering number* of $\mathcal{F}$ is

$$N(\mathcal{F}, p, \epsilon) := \inf\{|\mathcal{C}| : \mathcal{C} \text{ is an } \epsilon\text{-cover of } \mathcal{F}\}.$$

If $\mathcal{C} \subseteq \mathcal{F}$ is an $\epsilon$-*cover* of $\mathcal{F}$, then $\mathcal{C}$ is an $\epsilon$-*self cover* of $\mathcal{F}$. The $\epsilon$-*self-covering number* of $\mathcal{F}$ is

$$N^s(\mathcal{F}, p, \epsilon) := \inf\{|\mathcal{C}| : \mathcal{C} \text{ is an } \epsilon\text{-self-cover of } \mathcal{F}\}.$$

Clearly, $N^s(\mathcal{F}, p, \epsilon) \geq N(\mathcal{F}, p, \epsilon)$, and we establish the reverse relation.

**Lemma 12.** *For any $\epsilon \geq 0$, $N^s(\mathcal{F}, p, \epsilon) \leq N(\mathcal{F}, p, \epsilon/2)$.*

*Proof.* If $N(\mathcal{F}, p, \epsilon/2) = \infty$, the lemma clearly holds. Otherwise, let $\mathcal{C}$ be an $\epsilon/2$-cover of size $N(\mathcal{F}, p, \epsilon/2)$. We construct an $\epsilon$-self-cover of equal or smaller size.

For every subset $S_{\mathcal{C}} \in \mathcal{C}$, there is a subset $S = f(S_{\mathcal{C}}) \in \mathcal{F}$ with $p(S_{\mathcal{C}} \triangle f(S_{\mathcal{C}})) \leq \epsilon/2$. Otherwise, $S_{\mathcal{C}}$ could be removed from $\mathcal{C}$ to obtain a strictly smaller $\epsilon/2$ cover, which is impossible.

The collection $\{f(S_{\mathcal{C}}) : S_{\mathcal{C}} \in \mathcal{C}\} \subseteq \mathcal{F}$ has size $\leq |\mathcal{C}|$, and it is an $\epsilon$-self-cover of $\mathcal{F}$ because for any $S \in \mathcal{F}$, there is an $S_{\mathcal{C}} \in \mathcal{C}$ with $p(S \triangle S_{\mathcal{C}}) \leq \epsilon/2$, and by the triangle inequality, $p(S \triangle f(S_{\mathcal{C}})) \leq \epsilon$. ∎

Let $N_{\mathcal{F}, \epsilon} := \sup_p N(\mathcal{F}, p, \epsilon)$ and $N^s_{\mathcal{F}, \epsilon} := \sup_p N^s(\mathcal{F}, p, \epsilon)$ be the largest covering numbers under any distribution.

The next theorem bounds the covering number of $\mathcal{F}$ in terms of its VC-dimension.

**Theorem 13** ([VW96]). *There exists a universal constant $c$ such that for any $\epsilon > 0$, and any family $\mathcal{F}$ with VC dimension $V_{\mathcal{F}}$,*

$$N_{\mathcal{F}, \epsilon} \leq cV_{\mathcal{F}}\left(\frac{4e}{\epsilon}\right)^{V_{\mathcal{F}}}.$$

Combining the theorem and Lemma 12, we obtain the following corollary.

**Corollary 14.** *There exists a universal constant $c$ such that for any $\epsilon > 0$, and any family $\mathcal{F}$ with VC dimension $V_{\mathcal{F}}$,*

$$N^s_{\mathcal{F}, \epsilon} \leq cV_{\mathcal{F}}\left(\frac{8e}{\epsilon}\right)^{V_{\mathcal{F}}}.$$

The above corollary implies that for any distribution $p$, a VC class $\mathcal{F}$ has an $\epsilon$ self cover, under distribution $p$, of size $\mathcal{O}\left(V_{\mathcal{F}}\left(\frac{8e}{\epsilon}\right)^{V_{\mathcal{F}}}\right)$.

## C  A framework for distribution estimation from corrupted sample batches

We develop a general framework for learning distributions in $\mathcal{F}$ distance, leading to Theorem 1. Building on this framework, we derive a computationally efficient algorithm for learning in $\mathcal{F}_k$ distance, yielding Theorem 2.

Recall that the $\mathcal{F}$ distance between two distributions $p$ and $q$ is
$$||p - q||_{\mathcal{F}} = \sup_{S \in \mathcal{F}} |p(S) - q(S)|.$$

Our goal is to estimate $p$ to $\mathcal{F}$-distance $\mathcal{O}(\Delta)$, where $\Delta = \mathcal{O}\big(\beta\sqrt{\frac{\ln(1/\beta)}{n}}\big)$ is essentially the lower bound.

At a high level, the filtering algorithm removes the adversarial, or "outlier" batches, and returns a sub-collection $B' \subseteq B$ of batches whose empirical distribution $\bar{p}_{B'}$ is close to $p$ in $\mathcal{F}$ distance. The uniform deviation inequality in VC theory states that the sub-collection $B_G$ of good batches has empirical distribution $\bar{p}_{B_G}$ that approximates $p$ in $\mathcal{F}$ distance, thereby ensuring the existence of such a sub-collection when the number of batches $m$ is sufficiently large.

[JO19] developed a filtering algorithm for learning in TV-distance for a finite domain $\Omega = [k]$. The main drawback of this approach is that applying filtering algorithm directly for $\Sigma$-distance requires a number of samples linear in domain size, which is prohibitive for non-finite domains. Here we focus on general domains $\Omega$ and any collection of its subsets that has a finite VC-dimension.

Subsection C.1 describes certain filtration properties for a subset of $\Omega$ and using the subset that has these filtration properties as a filter. This can be viewed as a reinterpretation of the similar properties used in the filtering algorithm of [JO19]. Subsection C.2 uses these properties to develop a filtering algorithm for any finite collection of subsets. Subsection C.3 proves a Robust covering theorem to extends the filtering algorithm to VC family of subsets and proves Theorem 1. Subsection C.4 gives a computationally efficient filtering algorithm for the collection of subsets generated by a finite partition of the domain. Building on this, the next subsection C.5 gives an efficient algorithm for learning in $\mathcal{F}_k$ distance and proves Theorem 2.

## C.1 Using subsets as filters

We discuss how a subset $S \in \Sigma$ can be used as a filter. For this section, we fix a subset $S \in \Sigma$.

We show that if empirical estimates $\bar{\mu}_b(S)$ that batches $b \in B$ assigns to this subset $S$ satisfy certain properties then we can accurately learn its probability and use this subset as a filter. The following discussion develops some notation and intuitions that lead to these properties.

We start with the following observation. For every good batch $b \in B_G$, the empirical estimate $n \cdot \bar{\mu}_b(S)$ has a binomial distribution $\mathrm{Bin}(p(S), n)$, which implies that $\bar{\mu}_b(S)$ has a sub-gaussian distribution $\mathrm{subG}(p(S), \frac{1}{4n})$ with variance $\mathrm{V}(p(S))$. Hence, the empirical mean and variance of $\bar{\mu}_b(S)$ over $b \in B_G$ converges to the expected values $p(S)$ and $\mathrm{V}(p(S))$, respectively. Moreover, sub-gaussian property of the distribution of $\bar{\mu}_b(S)$ implies that, most of the good batches $b \in B_G$ assign the empirical probability $\bar{\mu}_b(S) \in p(S) \pm \tilde{O}(1/\sqrt{n})$.

In addition to the good batches, the collection $B$ of batches also includes an adversarial sub-collection $B_A$ of batches that constitute up to a $\beta-$fraction of $B$. If the difference between $p(S)$ and the average of $\bar{\mu}_b(S)$ over all adversarial batches $b \in B_A$ is $\leq \tilde{O}(\frac{1}{\sqrt{n}})$, namely comparable to the standard deviation of $\bar{\mu}_b(S)$ for the good batches $b \in B_G$, then the adversarial batches can change the overall mean of empirical probabilities $\bar{\mu}_b(S)$ by at most $\tilde{O}(\frac{\beta}{\sqrt{n}})$, which is within our tolerance. Hence, the mean of $\bar{\mu}_b(S)$ will deviate significantly from $p(S)$ only in the presence of a large number of adversarial batches $b \in B_A$ whose empirical probability $\bar{\mu}_b(S)$ differs from $p(S)$ by $\gg \tilde{\Omega}(\frac{1}{\sqrt{n}})$.

To quantify this effect, for a subset $S \in \Sigma$, let
$$\mathrm{med}(\bar{\mu}(S)) := \mathrm{median}\{\bar{\mu}_b(S) : b \in B\}$$
be the median empirical probability of $S$ over all batches. Property 1 (defined later) shows that w.h.p., the absolute difference between $\mathrm{med}(\bar{\mu}(S))$ and $p(S)$ is $\leq \mathcal{O}(1/\sqrt{n})$. The *corruption score* of batch $b$ for $S$ is

$$\psi_b(S) := \begin{cases} 0 & \text{if } |\bar{\mu}_b(S) - \mathrm{med}(\bar{\mu}(S))| \leq \mathcal{O}\Big(\sqrt{\frac{\ln(1/\beta)}{n}}\Big), \\ (\bar{\mu}_b(S) - \mathrm{med}(\bar{\mu}(S)))^2 & \text{otherwise.} \end{cases}$$

The preceding discussion shows that the corruption score of most good batches for the subset $S$ is zero and that adversarial batches that may significantly change the overall mean of empirical probabilities have high corruption score.

The *corruption score* of a sub-collection $B' \subseteq B$ for a subset $S$ is the sum of the *corruption score* of its batches,

$$\psi_{B'}(S) := \sum_{b \in B'} \psi_b(S).$$

A high corruption score of $B'$ for a subset $S$ indicates that $B'$ has many batches $b$ with large difference $|\bar{\mu}_b(S) - \mathrm{med}(\bar{\mu}(S))|$.

Next, we describe some essential properties that allows to a use subset $S$ as a filter. We later show that regardless of the samples in adversarial batches, with high probability, the empirical estimates $\bar{\mu}_b(S)$ for $b \in B$ satisfies the following four *filtration properties*.

1. The median of the estimates $\{\bar{\mu}_b(S) : b \in B\}$ is close to $p(S)$,

$$|\mathrm{med}(\bar{\mu}(S)) - p(S)| \leq \mathcal{O}(1/\sqrt{n}).$$

2. For every sub-collection $B'_G \subseteq B_G$ containing a large portion of the good batches, $|B'_G| \geq (1 - \beta/6)|B_G|$, the empirical mean of $\bar{\mu}_b(S)$ estimate $p(S)$ well,

$$|\bar{p}_{B'_G}(S) - p(S)| \leq \mathcal{O}\Big(\beta\sqrt{\frac{\ln(1/\beta)}{n}}\Big) = \mathcal{O}(\Delta),$$

3. The corruption score of the collection $B_G$ of good batches for subset $S$ is small,

$$\psi_{B'}(S) \leq \kappa_G := \mathcal{O}\Big(\frac{\beta m \ln(1/\beta)}{n}\Big).$$

4. For every sub-collection $B'_G \subseteq B_G$ s.t. $|B'_G| \geq (1 - \beta/6)|B_G|$, the empirical variance of $\bar{\mu}_b(S)$ estimate $\mathrm{V}(p(S))$ well,

$$\Big|\frac{1}{|B'_G|} \sum_{b \in B'_G} (\bar{\mu}_b(S) - p(S))^2 - \mathrm{V}(p(S))\Big| \leq \mathcal{O}\Big(\frac{\beta \ln(1/\beta)}{n}\Big).$$

If any of the four filtration properties holds for subset $S$, we say that $S$ has that particular property.

Next we show how a subset $S$ with the first three of the filtration properties, can be used as a filter. The last filtration property will be used later for deriving computationally efficient algorithms.

For subset $S$ that has filtration properties and for every sub-collection $B' \subseteq B$ that contain most good batches, the next lemma upper bounds the absolute difference between $p(S)$ and the empirical estimate $\bar{p}_{B'}(S)$ of the batches in $B'$ in terms of the corruption score of $B'$.

**Lemma 15.** *If subset $S$ has filtration properties 1- 3, then for any $B'$ such that $|B' \cap B_G| \geq (1 - \frac{\beta}{6})|B_G|$ such that $\psi_{B'}(S) \leq t \cdot \kappa_G$, for some $t \geq 0$, then*

$$|\bar{p}_{B'}(S) - p(S)| \leq \mathcal{O}\Big((\sqrt{t} + 1)\Delta\Big).$$

The lemma is related to Lemma 4 in [JO19], hence we provide only a high-level argument. For any sub-collection $B'$ retaining a major portion of good batches, from filtration property 2, the mean of $\bar{\mu}_b(S)$ of the good batches $B' \cap B_G$ approximates $p(S)$. Showing that a small corruption score of $B'$ implies that the adversarial batches $B' \cap B_A$ have limited effect on $\bar{p}_{B'}(S)$ proves the lemma.

Next, we describe the *Batch-Deletion* algorithm of [JO19] and its performance guarantees.

Given a subset $S$ with filtration property 3 and any sub-collection $B'$, the algorithm successively removes batches from $B'$, ensuring that each batch removed is adversarial with high probability. The algorithm stops deleting batches when the corruption score of the remaining sub-collection for $S$ is small.

---

**Algorithm 1** Batch-Deletion

---

1: **Input:** Sub-Collection $B'$ of Batches, subset $S$, med=med($\bar{\mu}(S)$), and $\kappa_G$
2: **Output:** A smaller sub-collection $B'$ of batches
3: **Comment:** The terms $\kappa_G$, $\psi_b(S)$, and $\psi_{B'}(S)$ used below are defined earlier in this section, and computing $\psi_b(S)$ and $\psi_{B'}(S)$ require med($\bar{\mu}(S)$) as input (that depends on all batches $B$).
4: **while** $\psi_{B'}(S) \geq 20\kappa_G$ **do**
5:     Select a single batch $b \in B'$ where batch $b$ is selected with probability $\frac{\psi_b(S)}{\psi_{B'}(S)}$;
6:     $B' \leftarrow \{B' \setminus b\}$;
7: **end while**
8: **return** $(B')$;

---

The next lemma, characterizes the performance of the Batch-Deletion algorithm.

**Lemma 16.** *Let $B' \subseteq B$ and subset $S$ be the input of the Batch-Deletion algorithm. If subset $S$ has filtration property 3, then:*

1. *Each batch that gets removed from $B'$ by Batch-Deletion algorithm is an adversarial batch with probability $\geq 0.95$.*

2. *Batch-Deletion returns updated sub-collection $B'$ such that $\psi_{B'}(S) < 20\kappa_G$.*

*Proof.* The first statement in the lemma follows as

$$\Pr[\text{Deleting a batch from } B_G \cap B'] = \sum_{b \in B' \cap B_G} \frac{\psi_b(S)}{\psi_{B'}(S)} \leq \frac{\sum_{b \in B_G} \psi_b(S)}{\psi_{B'}(S)} \leq \frac{\kappa_G}{20\kappa_G} \leq 0.05,$$

here we used filtration property 3. The second statement in the Lemma follows from step 4 of Batch-Deletion algorithm. ∎

Lemma 15 implies that if a sub-collection $B'$ has most of the good batches and has a small corruption score for subset $S$, then $\bar{\mu}_b(S)$ is close to $p(S)$.

Lemma 16 implies that if sub-collection $B'$ has large corruption for subset $S$, then there is a probabilistic method that removes more adversarial batches from $B'$ then good batches and lowers the corruption.

The next subsection builds on these two Lemma and gives a simple filtering algorithm for any finite collection of subsets $C \subseteq \Sigma$ whose subsets $S \in C$ has filtration properties 1-3.

### C.2 Filtering algorithms for finite collection of subsets

Given any finite collection of subsets $C \subseteq \mathcal{F}'$, algorithm 2, described next, uses the Batch-Deletion algorithm to successively update $B$ and decrease the corruption score for each subset $S \in C$.

---

**Algorithm 2** Filtering Algorithm

---

1: **Input:** Collection $B$ of Batches, finite subset family $C \subseteq \Sigma$, adversarial batches fraction $\beta$
2: **Output:** A sub-collection $B^*$ of batches.
3: **Comment:** The terms $\kappa_G$, $\psi_{B'}(S)$, and med($\bar{\mu}(S)$) used below are defined earlier in this section
4: $B' = B$;
5: **for** $S \in C$ **do**
6:     **if** $\psi_{B'}(S) \geq 20\kappa_G$ **then**
7:         med $\leftarrow$ med($\bar{\mu}(S)$);
8:         $B' \leftarrow$ Batch-Deletion($B', S,$ med);
9:     **end if**
10: **end for**
11: $B^* \leftarrow B'$
12: **return** $(B^*)$;

---

The next lemma characterizes the algorithm's performance.

**Lemma 17.** *Let $C \subseteq \Sigma$ be a finite collection of subsets. If all subsets in $C$ have filtration properties 1, 2 and 3, then algorithm 2 returns a sub-collection of batches $B^*$ such that with probability $\geq 1 - e^{-O(\beta m)}$, $|B^* \cap B_G| \geq (1 - \frac{\beta}{6})|B_G|$ and*

$$||p - p_{B^*}||_C = \max_{S \in C} |p(S) - p_{B^*}(S)| \leq \mathcal{O}(\Delta).$$

The proof of the lemma is immediate from Lemmas 15 and 16.

We note that $|B^*| \geq (1 - \frac{\beta}{6})|B_G| \geq (1 - \frac{\beta}{6})(1 - \beta)m > m/2$, as $\beta \in (0, 0.4]$. Therefore, w.h.p. $B^*$ retains at least half of the overall batches.

## C.3 Robust covering theorem for learning in $\mathcal{F}$ distance and Proof of Theorem 1

A subset family $\mathcal{F}$, with finite VC dimension, can have potentially uncountable subsets, hence, even if all subsets in $\mathcal{F}$ have filtration properties 1-3, we may not be able to use filtering algorithm directly for subset family $\mathcal{F}$. The *Robust covering* theorem proved here overcomes this challenge.

Recall that the collection $B$ includes adversarial batches that can cause the empirical distribution of all batches $\bar{p}_B$ to be at an $\mathcal{F}$-distance $\mathcal{O}(\beta)$ from $p$.

Yet for any $\epsilon > 0$, any sub-collection $B' \subseteq B$ consisting of at least half of the batches, and for any $\epsilon$-cover $C$ of $\mathcal{F}$ under the empirical distribution $\bar{p}_B$ of all batches $B$, the next theorem upper bounds, $||\bar{p}_{B'} - p||_{\mathcal{F}}$, the $\mathcal{F}$-distance between $p$ and the empirical distribution induced by $B'$ in terms of $||\bar{p}_{B'} - p||_C$, the $C$-distance between them.

Let $\mathcal{G}$ be a VC-class of subsets such that $\mathcal{F} \subseteq \mathcal{G}$. The theorem allows the $\epsilon$-cover $C$ of $\mathcal{F}$ to include subsets from a larger class of subsets $\mathcal{G}$. Although, one can always choose a cover of $\mathcal{F}$ from within the class, as we will see in later subsections, for computationally efficient algorithms some additional structure in the cover may be desired. And to choose such a cover, we will choose its elements (subsets) from a larger class of subsets than $\mathcal{F}$.

**Theorem 18** (Robust covering). *For any $\epsilon > 0$, any subset family $\mathcal{G} \supseteq \mathcal{F}$ with VC dimension $V_{\mathcal{G}}$, and $m \cdot n \geq \mathcal{O}(\frac{V_{\mathcal{G}} \log(1/\epsilon) + \log(1/\delta)}{\epsilon^2})$, let $C \subseteq \mathcal{G}$ be an $\epsilon$-cover of family $\mathcal{F}$ under the empirical distribution $\bar{p}_B$. With probability $\geq 1 - \delta$, for every sub-collection of batches $B' \subseteq B$ of size $|B'| \geq m/2$,*

$$||\bar{p}_{B'} - p||_{\mathcal{F}} \leq ||\bar{p}_{B'} - p||_C + 5\epsilon.$$

*Proof.* Consider any batch sub-collection $B' \subseteq B$. For every $S, S' \in \Sigma$, by the triangle inequality,

$$|\bar{p}_{B'}(S) - p(S)| = \left| \left( \bar{p}_{B'}(S') + \bar{p}_{B'}(S \setminus S') - \bar{p}_{B'}(S' \setminus S) \right) - \left( p(S') + p(S \setminus S') - p(S' \setminus S) \right) \right|$$
$$\leq |\bar{p}_{B'}(S') - p(S')| + \bar{p}_{B'}(S \setminus S') + \bar{p}_{B'}(S' \setminus S) + p(S \setminus S') + p(S' \setminus S)$$
$$= |\bar{p}_{B'}(S') - p(S')| + \bar{p}_{B'}(S \triangle S') + p(S \triangle S'). \tag{4}$$

Since $C$ is an $\epsilon$-cover under $\bar{p}_B$, for every $S \in \mathcal{F}$ there is an $S' \in C$ such that $\bar{p}_B(S \triangle S') \leq \epsilon$. For such pairs, we bound the second term on the right in the above equation.

$$\bar{p}_{B'}(S \triangle S') = \frac{1}{|B'|n} \sum_{b \in B'} \sum_{i \in [n]} \mathbf{1}_{S \triangle S'}(X_i^b)$$
$$\leq \frac{1}{|B'|n} \sum_{b \in B} \sum_{i \in [n]} \mathbf{1}_{S \triangle S'}(X_i^b)$$
$$= \frac{|B|}{|B'|} \cdot \frac{1}{|B|n} \sum_{b \in B} \sum_{i \in [n]} \mathbf{1}_{S \triangle S'}(X_i^b)$$
$$= \frac{m}{|B'|} \bar{p}_B(S \triangle S') \leq \frac{m\epsilon}{|B'|}. \tag{5}$$

Choosing $B' = B_G$ in the above equation and using $B_G = (1 - \beta)m \geq m/2$ gives,

$$\bar{p}_{B_G}(S \triangle S') < 2\epsilon. \tag{6}$$

Then

$$p(S \triangle S') \leq |p(S \triangle S') - \bar{p}_{B_G}(S \triangle S')| + \bar{p}_{B_G}(S \triangle S')$$

$$\overset{(a)}{\leq} \sup_{S,\, S' \in \mathcal{G}} |p(S \triangle S') - \bar{p}_{B_G}(S \triangle S')| + 2\epsilon$$

$$\overset{(b)}{\leq} \epsilon + 2\epsilon,$$

with probability $\geq 1 - \delta$, here (a) used the fact that $\mathcal{C}, \mathcal{F} \subseteq \mathcal{G}$ and equation (6) and (b) follows from Lemma 24. Combining equations (4), (5) and the above equation completes the proof. ∎

In contrast to the class $\mathcal{F}$, which could be infinite, we can always choose a cover $\mathcal{C}$ of finite size and therefore run filtering algorithm 2 for $C = \mathcal{C}$ to learn in $\mathcal{C}$ distance. Robust covering theorem implies that if $\mathcal{C}$ is $\epsilon$-cover of family $\mathcal{F}$, under distribution $\bar{p}_B$, where $\epsilon = \mathcal{O}(\Delta)$, then for learning in $\mathcal{C}$ distance suffices to learn in $\mathcal{F}$ distance.

The only step that remains is to find a cover whose subsets have filtration properties. The next lemma establishes that every subsets in a given VC-subset family $\mathcal{G}$ has filtration properties.

**Lemma 19.** *For any given subset family $\mathcal{G}$ with finite VC dimension and the number of batches $m \geq \mathcal{O}(\frac{V_{\mathcal{G}} \log(n/\beta) + \log(1/\delta)}{\beta^2})$. With probability $\geq 1 - \delta$, all subsets in $\mathcal{G}$ has filtration properties 1- 4.*

The proof of the lemma appears in section D.

Note that the number of samples required in the lemma increase with the VC-complexity of $\mathcal{G}$. Therefore, to obtain sample optimal algorithm, we choose $\mathcal{G} = \mathcal{F}$, and $\mathcal{C}$ to be any finite $\epsilon$-self-cover of $\mathcal{F}$ under distribution $\bar{p}_B$, where $\epsilon \leq \mathcal{O}(\Delta)$. The existence of such a self-cover is guaranteed by Corollary 14.

The above lemma implies that w.h.p. all subsets in $C$ has filtration property. Therefore, we run algorithm 2 for $C = \mathcal{C}$. Then combining Lemma 17 and robust covering theorem 18 implies learning in $\mathcal{F}$ distance and gives Theorem 1.

**Theorem 20** (Theorem 1 restated). *For any $\beta \leq 0.4$, $\delta > 0$, $\mathcal{F}$, and $m \cdot n \geq \mathcal{O}\left(\frac{V_{\mathcal{F}} \log(1/\Delta) + \log 1/\delta}{\Delta^2} \cdot \log(\frac{1}{\beta})\right)$, there is a non-constructive algorithm that with probability $\geq 1 - \delta$ returns a sub-collection of batches $B^*$ such that $|B^* \cap B_G| \geq (1 - \frac{\beta}{6})|B_G|$ and*

$$\|p - \bar{p}_{B^*}\|_{\mathcal{F}} \leq \mathcal{O}(\Delta).$$

## C.4 Computationally efficient algorithm for subsets generated by a partition

For estimating $p$ in $\mathcal{F}$-distance, in the previous subsection, we chose $C$ to be a cover of $\mathcal{F}$ and estimated $p$ in $C$ distance. Then to estimate $p$ in $C$ distance, algorithm 2 iterates through all subsets in $C$ one by one, and therefore, has run-time at least linear in the size of the subset family $C$. But the size of the cover of $\mathcal{F}$ may grow exponentially with the VC-dimension of family $\mathcal{F}$. This makes the algorithm 2 computationally prohibitive even for subset family $\mathcal{F}$ with moderate VC-dimension. Here we show that if subset collection $C$ has a certain structure then this time complexity can be reduced significantly.

For any $\ell > 0$, we consider $C$ which is the collection of all subsets generated by an $\ell$-partition of the domain $\Omega$. Here we give a filtering algorithm that has run time only polynomial in $\ell$, whereas the size of subset collection $C$ is $2^{\ell}$.

For any integer $\ell > 0$, let $\xi : \Omega \to [\ell]$ be any function. This function $\xi$ partitions the domain $\Omega$ into $\ell$ disjoint parts. For $j \in [\ell]$, let $\xi_j := \xi^{-1}(j)$ denote the $j^{th}$ *partition element* in the partition created by $\xi$. Clearly the partition elements $\xi_j$'s are disjoint and their union is $\Omega$. We refer to $\xi$ as *partition function*. Note that a partition function $\xi$ is uniquely determined by the corresponding partition elements $\xi_j$'s.

For a subset $D \subseteq [\ell]$, let

$$S_D^{\xi} := \cup_{j \in D} \xi_j,$$

be the union of the partition elements $\xi_j$'s corresponding to the elements of $D$. Define the collection of subsets

$$C^{\xi} := \{S_D^{\xi} : D \in 2^{[\ell]}\}$$

to be the family of all possible unions of $\xi_j$'s. Clearly, $|C^\xi| = 2^\ell$.

We show that if all subsets $S \in C^\xi$ have filtration properties 1- 4, then $p$ can be estimated to a small $C^\xi$-distance in time polynomial in $\ell$ rather than exponential.

For finite domain $\Omega' = [\ell]$, [JO19] derived a method that for any batch sub-collection $B'$, containing a majority of good batches, can find a subset in $2^{[\ell]}$ for which the corruption score of $B'$ is within a constant times the maximum in time only polynomial in the domain size $\ell$, when all subsets in $2^{[\ell]}$ have filtration properties 1- 4. Then instead of iterating over all $2^\ell$ subsets, as in algorithm 2, they find the subsets with high corruption score efficiently and use the Batch Deletion procedure for these subsets. This leads to a computationally efficient algorithm for learning discrete distributions $p$.

To obtain a computationally efficient algorithm for learning in $C^\xi$ distance, we first reduce this problem to that of robustly learning distributions over finite domains in total variation distance and then use the algorithm in [JO19].

**Theorem 21.** *Let $\xi : \Omega \to [\ell]$ be any partition function and let $C^\xi$ be the collection of all possible unions of the partition elements $\xi_j$'s. If all subsets in $C^\xi$ have filtration properties 1- 4, then there is an algorithm that runs in time polynomial in all parameters $\ell$, $m$, and $n$, and with probability $\geq 1 - e^{-O(\beta m)}$ returns a sub-collection of batches $B^* \subseteq B$ such that $|B^* \cap B_G| \geq (1 - \beta/6)|B_G|$ and*

$$||p - \bar{p}_{B^*}||_{C^\xi} \leq \mathcal{O}(\Delta).$$

*Proof.* First note that $\xi$ transforms any distribution $q$ over $\Omega$ to the discrete distribution $q^\xi$ over $\Omega' = [\ell]$, where $q^\xi(j) := q(\xi_j)$ for each $j \in [\ell]$. Recall that any subset $D \subseteq [\ell]$, corresponds one to one with a subset $S_D^\xi = \cup_{j \in D} \xi_j$ in $C^\xi$. It follows that for any distribution $q$ over $\Omega$, and $D \subseteq [\ell]$,

$$q(S_D^\xi) = q^\xi(D).$$

Recall that $\bar{p}_{B'}$ denotes the empirical distribution induced by a sub-collection $B'$, therefore $\bar{p}_{B'}^\xi$ denotes the empirical distribution induced by a sub-collection $B'$ over the transformed domain $[\ell]$.

From the one-to-one correspondence between subsets in $C^\xi$ and subsets in $2^{[\ell]}$ it follows that all subsets in $C^\xi$ have filtration properties iff all subsets in $2^{[\ell]}$ have filtration properties for the transformed distributions $p^\xi$ and transformed empirical distribution of the sample batches.

Theorem 9 in [JO19] implies that, if all subsets in $2^{[\ell]}$ have filtration properties 1- 4 then algorithm 2 therein runs in time polynomial in the domain size $\ell$, the number of batches $m$, and the batch-size $n$, and with probability $\geq 1 - e^{-O(\beta m)}$ returns a sub-collection of batches $B^* \subseteq B$ such that $|B^* \cap B_G| \geq (1 - \beta/6)|B_G|$ and

$$||p^\xi - \bar{p}_{B^*}^\xi||_{TV} \leq \mathcal{O}(\Delta).$$

Next, for any pair of distributions $q_1$ and $q_2$ over the domain $\Omega$, we show that $C^\xi$-distance between them is the same as the total variation distance between $q_1^\xi$ and $q_2^\xi$. For every distribution pair $q_1, q_2$ over $\Omega$,

$$\begin{aligned}
||q_1 - q_2||_{C^\xi} &= \max_{S \in C^\xi} |q_1(S) - q_2(S)| \\
&= \max_{S_D^\xi \in C^\xi} |q_1(S_D^\xi) - q_2(S_D^\xi)| \\
&= \max_{D \in 2^{[\ell]}} |q_1^\xi(D) - q_2^\xi(D)| \\
&= ||q_1^\xi - q_2^\xi||_{TV}.
\end{aligned}$$

Therefore,

$$||p - \bar{p}_{B^*}||_{C^\xi} = ||p^\xi - \bar{p}_{B^*}^\xi||_{TV} \leq \mathcal{O}(\Delta). \qquad \blacksquare$$

### C.5 Computationally efficient algorithm for learning in $\mathcal{F}_k$ distance and proof of Theorem 2

Recall that $\mathcal{F}_k$ is the collection of all unions of at most $k$ intervals over $\mathbb{R}$.

In the previous subsection we showed that for a partition function $\xi$, we can learn in $C^\xi$-distance efficiently. To obtain a computationally efficient algorithm for learning in $\mathcal{F}_k$ distance, we give a partition function $\xi^* : \mathbb{R} \to [\ell]$, for an appropriate $\ell$ to be chosen later, such that the collection of subsets $C^{\xi^*}$ forms an $\epsilon$-cover of $\mathcal{F}_k$ under the empirical distribution $\bar{p}_B$.

Recall that $B$ is a collection of $m$ batches and each batch has $n$ samples. Let $s = n \cdot m$ and let $x^s = x_1, x_2, \ldots, x_s \in \mathbb{R}$ be the samples of $B$ arranged in non-decreasing order. And recall that the points $x^s$ induce an empirical measure $\bar{p}_B$ over $\mathbb{R}$, where for $S \subseteq \mathbb{R}$,

$$\bar{p}_B(S) = |\{i : x_i \in S\}|/s.$$

Let $t := \frac{s}{\ell}$, and for simplicity assume that it is an integer. Recall that a partition function $\xi$ is uniquely determined by the corresponding partition elements $\xi_j$'s. Let $\xi^* : \mathbb{R} \to [\ell]$ be the partition function with partition elements $\{\xi_1^*, \ldots, \xi_\ell^*\}$ of $\mathbb{R}$, where

$$\xi_j^* := \begin{cases} (-\infty, x_t] & j = 1, \\ (x_{(j-1)t}, x_{jt}] & 2 \le j < \ell, \\ (x_{s-t}, \infty) & j = \ell. \end{cases}$$

Note that all elements of the partition $\{\xi_1^*, \ldots, \xi_\ell^*\}$ are intervals of $\mathbb{R}$. Recall that $C^{\xi^*}$ is is formed by all possible unions of these $\ell$ intervals. Clearly $C^{\xi^*} \subseteq \mathcal{F}_\ell$, as $\mathcal{F}_\ell$ contains all unions of $\ell$ intervals over $\mathbb{R}$.

We show that $C^{\xi^*}$ is an $2k/\ell$−cover of $\mathcal{F}_k$ under the empirical distribution $\bar{p}_B$ of points $x_1^s$.

**Lemma 22.** *For any $k$, and $\ell$, $C^{\xi^*}$ is a $\frac{2k}{\ell}$-cover of $\mathcal{F}_k$ under $\bar{p}_B$.*

*Proof.* Any set $S \in \mathcal{F}_k$ is a union of $k$ real intervals $I_1 \cup I_2 \cup \ldots \cup I_k$. Let $S^* \subseteq \mathbb{R}$ be the union of all $\xi_j^*$-partition elements (intervals) that are fully contained in one of the intervals $I_1, \ldots, I_k$. By definition, $S^* \in C^\xi$, and we show that $\bar{p}_B(S \triangle S^*) \le 2k/\ell$. By construction, $S^* \subseteq S$, hence,

$$\bar{p}_B(S \triangle S^*) = \bar{p}_B(S \setminus S^*) = \sum_{j=1}^k \bar{p}_B(I_j \setminus S^*) = \sum_{j=1}^k \frac{|\{x_i \in I_j \setminus S^*\}|}{s} \le \sum_{j=1}^k 2 \cdot \frac{t}{s} = \frac{2k}{\ell},$$

where the inequality follows as each $I_j \setminus S^*$ contains at most $t$ points and the left and right. ∎

Next choose $\ell = \frac{2k}{\epsilon}$ then the lemma implies that the corresponding $C^{\xi^*}$ is an $\epsilon$-cover of $\mathcal{F}_k$ under $\bar{p}_B$. As discussed earlier $C^{\xi^*} \subseteq \mathcal{F}_\ell$. Then choosing $\mathcal{G} = \mathcal{F}_\ell$ in Lemma 19 implies that $w.h.p.$ all subsets in $C^{\xi^*}$ has filtering properties. Then combining Theorem 21 and robust covering theorem 18, and choosing $\epsilon = \mathcal{O}(\Delta)$, we get the following theorem that implies learning in $\mathcal{F}_k$ distance.

We note that this computationally efficient algorithm uses $\mathcal{O}(1/\Delta)$ times more sample than information theoretic algorithm in section C.3, because here we chose the cover of $\mathcal{F}_k$ from the class $\mathcal{G} = \mathcal{F}_{k/\Delta}$. And $\mathcal{F}_{k/\Delta}$ has VC dimension $\mathcal{O}(k/\Delta)$, which is $\mathcal{O}(1/\Delta)$ times the VC-dimension of the class $\mathcal{F}_k$.

**Theorem 23** (Theorem 2 restated)**.** *For any given $\beta \le 0.4$, $\delta > 0$, $k > 0$, and $m \cdot n \ge \mathcal{O}\left(\frac{k \log(1/\Delta) + \log 1/\delta}{\Delta^3} \cdot \log(\frac{1}{\beta})\right)$, there is an algorithm that runs in time polynomial in all parameters, and with probability $\ge 1 - \delta$ returns a sub-collection of batches $B^*$ such that $|B^* \cap B_G| \ge (1 - \frac{\beta}{6})|B_G|$ and*

$$||\bar{p}_{B^*} - p||_{\mathcal{F}_k} \le \mathcal{O}(\Delta).$$

# D   Properties of the Collection of Good Batches

**Lemma 24.** *Let $\mathcal{G}$ be a VC family of subsets of $\Omega$. Then for any $\delta > 0$ and $|B_G| \cdot n \ge \mathcal{O}\left(\frac{V_\mathcal{G} \log(1/\epsilon) + \log(1/\delta)}{\epsilon^2}\right)$, with probability $\ge 1 - \delta$,*

$$\sup_{S, S' \in \mathcal{G}} \max \left\{ \frac{\bar{p}_{B_G}(S \triangle S') - p(S \triangle S')}{\sqrt{\bar{p}_{B_G}(S \triangle S')}}, \frac{p(S \triangle S') - \bar{p}_{B_G}(S \triangle S')}{\sqrt{p(S \triangle S')}} \right\} \le \epsilon.$$

*Proof.* Consider the collection of symmetric differences of subsets in $\mathcal{G}$,

$$\mathcal{G}_\triangle := \{S \triangle S' : S, S' \in \mathcal{G}\}.$$

The next auxiliary lemma bounds the shatter coefficient of $\mathcal{G}_\triangle$.

**Lemma 25.** *For $t \geq V_\mathcal{G}$, $S_{\mathcal{G}_\triangle}(t) \leq \left(\frac{t e}{V_\mathcal{G}}\right)^{2V_\mathcal{G}}$.*

*Proof.* For $t \geq V_\mathcal{G}$ and $x_1, x_2, .., x_t \in \Omega$, let

$$\mathcal{G}(x_1^t) = \{\{x_1, x_2, .., x_t\} \cap S : S \in \mathcal{G}\}.$$

Note that $S_\mathcal{G}(t) = \max_{x_1, \ldots, x_t} |\mathcal{G}(x_1^t)|$.

From the definition of shatter coefficient $|\mathcal{G}(x_1^t)| \leq S_\mathcal{G}(t)$. Then

$$|\mathcal{G}_\triangle(x_1^t)| = |\{\{x_1, \ldots, x_t\} \triangle \{x_1', \ldots, x_t'\} : S, S' \in \mathcal{G}(x_1^t)\}| \leq (S_\mathcal{G}(t))^2 \leq \left(\frac{t e}{V_\mathcal{G}}\right)^{2V_\mathcal{G}}. \qquad \blacksquare$$

Applying Theorem 11 for family of subsets $\mathcal{G}_\triangle$, and using Lemma 25, for $|B_G| \cdot n \geq \mathcal{O}(\frac{V_\mathcal{G} \log(1/\epsilon) + \log(1/\delta)}{\epsilon^2})$, with probability $\geq 1 - \delta$,

$$\sup_{S \in \mathcal{G}_\triangle} \max \left\{ \frac{\bar{p}_{B_G}(S) - p(S)}{\sqrt{\bar{p}_{B_G}(S)}}, \sup_{S \in \mathcal{G}} \frac{p(S) - \bar{p}_{B_G}(S)}{\sqrt{p(S)}} \right\} \leq \epsilon. \qquad \blacksquare$$

## D.1 Proof of Lemma 19

First we list some auxiliary properties for a subset $S$, each of which is either one of the filtration property or helps in deriving one of the filtration property.

**(i)** For every $B_G' \subseteq B_G$, such that $|B_G'| \geq (1 - \beta/6)|B_G|$

$$|\bar{p}_{B_G'}(S) - p(S)| \leq \mathcal{O}\left(\beta \sqrt{\frac{\ln(1/\beta)}{n}}\right).$$

**(ii)** For every $B_G' \subseteq B_G$, such that $|B_G'| \geq (1 - \beta/6)|B_G|$

$$\left| \frac{1}{|B_G'|} \sum_{b \in B_G'} (\bar{\mu}_b(S) - p(S))^2 - \mathrm{V}(p(S')) \right| \leq \mathcal{O}\left(\frac{\beta \ln(\frac{1}{\beta})}{n}\right).$$

**(iii)**

$$\left|\{b \in B_G : |\bar{\mu}_b(S) - p(S)| \geq \mathcal{O}\left(\sqrt{\frac{\ln(1/\beta)}{n}}\right)\}\right| \leq \mathcal{O}(\beta) \cdot |B_G|.$$

**(iv)**

$$\left|\{b \in B_G : |\bar{\mu}_b(S) - p(S)| \geq \mathcal{O}\left(\frac{1}{\sqrt{n}}\right)\}\right| \leq \mathcal{O}(1) \cdot |B_G|.$$

**(v)** For every $B_G' \subseteq B_G$, such that $|B_G'| \leq \mathcal{O}(\beta) \cdot |B_G|$

$$\sum_{b \in B_G'} (\bar{\mu}_b(S) - p(S))^2 < \mathcal{O}\left(\beta|B_G|\frac{\ln(1/\beta)}{n}\right),$$

The next lemma shows that these properties hold for a fix subset $S$.

**Lemma 26.** *For any given subset $S \in \Sigma$ and for $|B_G| \geq O(\frac{\log 1/\delta}{\beta^2 \ln(1/\beta)})$. With probability $\geq 1 - \delta$, subset $S$ has all auxiliary properties (i)–(v). Further, if these auxiliary properties hold for subset $S$ then subset $S$ has filtration properties 1- 4.*

The above Lemma, though not stated explicitly, is implied by Section A.1 and Section A.2 in [JO19]. In particular, the auxiliary properties **(i)** and **(ii)** are implied by Lemma 11, **(iii)** and **(iv)** are implied by Lemma 10, and **(v)** is implied by Lemma 12, and section A.2 therein showed that these auxiliary properties imply filtration properties 1- 4. Hence, we use the lemma without proving it again here.

Therefore, to prove Lemma 19, it suffices to show these auxiliary properties for subsets in $\mathcal{G}$.

The next Lemma extends the auxiliary properties to all subsets in given a VC class $\mathcal{G}$.

**Lemma 27.** *For any given subset family $\mathcal{G}$ with finite VC dimension and $|B_G| \geq O(\frac{V_{\mathcal{G}} \log(n/\beta) + \log 1/\delta}{\beta^2})$. With probability $\geq 1 - \delta$, all subsets in $\mathcal{G}$ has all auxiliary properties **(i)**–**(v)**.*

*Proof.* From Corollary 14, there exist a self $\epsilon$-cover $\mathcal{C}^*$ of $\mathcal{G}$ under the distribution $p$ of size $\mathcal{O}\big(V_{\mathcal{G}}(\frac{8e}{\epsilon})^{V_{\mathcal{G}}}\big)$. For this section, fix $\epsilon = \mathcal{O}(\frac{\beta^2}{n})$.

For any $S \in C^*$, for $|B_G| \geq O\Big(\frac{\log \frac{2|\mathcal{C}^*|}{\delta}}{\beta^2 \ln(1/\beta)}\Big) = O(\frac{V_{\mathcal{G}} \log(n/\beta) + \log 1/\delta}{\beta^2 \ln(1/\beta)})$, Lemma 26 implies that the auxiliary properties **(i)**–**(v)** with probability $\geq 1 - \frac{\delta}{2|\mathcal{C}^*|}$.

Therefore, taking the union bound over the complement, the auxiliary properties **(i)**–**(v)** hold for all subsets in $\mathcal{C}^*$ with probability $\geq 1 - \frac{\delta}{2}$.

Next, we extend these properties for all subsets in $\mathcal{G}$.

For subset $S \in \mathcal{G}$ choose $S' \in \mathcal{C}^*$ such that $p(S \triangle S') \leq \epsilon$. Existence of such a subset $S' \in \mathcal{C}^*$ is guaranteed for all $S \in \mathcal{G}$ as $\mathcal{C}^*$ is an $\epsilon-$cover under $p$. The properties for $S'$ holds, since it is a part of the cover $\mathcal{C}'$. To extend the auxiliary properties to all subsets in $\mathcal{G}$, we show that if the properties hold for $S'$, then they also hold for subset $S$.

Note that for any subset $S, S' \in \mathcal{G}$ with $p(S \triangle S') \leq \mathcal{O}(\frac{\beta^2}{n}) = \mathcal{O}(\epsilon)$.

For $|B_G| \cdot n \geq O(\frac{V_{\mathcal{G}} \log(n/\beta) + \log 1/\delta}{\beta^2} \cdot n)$, Lemma 24 implies that with probability $\geq 1 - \delta/2$

$$\bar{p}_{B_G}(S \triangle S') \leq \mathcal{O}(\frac{\beta^2}{n}) = \mathcal{O}(\epsilon). \tag{7}$$

For any batch $b \in B$

$$\bar{\mu}_b(S) - p(S) = \Big(\bar{\mu}_b(S') + \bar{\mu}_b(S \setminus S') - \bar{\mu}_b(S' \setminus S)\Big) - \Big(p(S') + p(S \setminus S') - p(S' \setminus S)\Big)$$
$$= \Big(\bar{\mu}_b(S') - p(S')\Big) + \Big(\bar{\mu}_b(S \setminus S') - \bar{\mu}_b(S' \setminus S)\Big) - \Big(p(S \setminus S') - p(S' \setminus S)\Big).$$

From the above equation, we get

$$\Big|\Big(\bar{\mu}_b(S) - p(S)\Big) - \Big(\bar{\mu}_b(S') - p(S')\Big)\Big| \leq \bar{\mu}_b(S \setminus S') + \bar{\mu}_b(S' \setminus S) + p(S \setminus S') + p(S' \setminus S)$$
$$= \bar{\mu}_b(S \triangle S') + p(S \triangle S')$$
$$\leq \bar{\mu}_b(S \triangle S') + \mathcal{O}(\epsilon). \tag{8}$$

Next, we extend property **(i)** to subset $S$.

$$|\bar{p}_{B'_G}(S) - p(S)| = \Big|\frac{1}{|B'_G|} \sum_{b \in B'_G} \bar{\mu}_b(S) - p(S)\Big| = \Big|\frac{1}{|B'_G|} \sum_{b \in B'_G} \Big(\bar{\mu}_b(S) - p(S)\Big)\Big|$$
$$\overset{(a)}{\leq} \Big|\frac{1}{|B'_G|} \sum_{b \in B'_G} \Big(\bar{\mu}_b(S') - p(S')\Big)\Big| + \Big|\frac{1}{|B'_G|} \sum_{b \in B'_G} \Big(\bar{\mu}_b(S \triangle S') + \mathcal{O}(\epsilon)\Big)\Big|$$
$$\leq \Big|\frac{1}{|B'_G|} \sum_{b \in B'_G} \bar{\mu}_b(S') - p(S')\Big| + \Big|\frac{1}{|B'_G|} \sum_{b \in B_G} \bar{\mu}_b(S \triangle S')\Big| + \mathcal{O}(\epsilon)$$
$$\leq |\bar{p}_{B'_G}(S') - p(S')| + \frac{|B_G|}{|B'_G|} \bar{p}_{B_G}(S \triangle S') + \mathcal{O}(\epsilon)$$
$$\overset{(b)}{\leq} \mathcal{O}\Big(\beta \sqrt{\frac{\ln(1/\beta)}{n}}\Big) + \frac{1}{(1 - \beta/6)} \cdot \mathcal{O}(\epsilon) + \mathcal{O}(\epsilon)$$

$$\leq \mathcal{O}\left(\beta\sqrt{\frac{\ln(1/\beta)}{n}}\right),$$

here (a) uses (8) and (b) uses that the property (i) holds for $S'$.

Next, we extend property (i) to subset $S$. From equation (8) we get

$$(\bar{\mu}_b(S) - p(S))^2 \leq \left(|\bar{\mu}_b(S') - p(S')| + (\bar{\mu}_b(S\triangle S') + \mathcal{O}(\epsilon))\right)^2$$
$$= (\bar{\mu}_b(S') - p(S'))^2 + 2|\bar{\mu}_b(S') - p(S')|(\bar{\mu}_b(S\triangle S') + \mathcal{O}(\epsilon)) + (\bar{\mu}_b(S\triangle S') + \mathcal{O}(\epsilon))^2.$$

Therefore,

$$\sum_{b\in B'_G} (\bar{\mu}_b(S) - p(S))^2 - \sum_{b\in B'_G} (\bar{\mu}_b(S') - p(S'))^2$$

$$\leq \sum_{b\in B'_G} 2|\bar{\mu}_b(S') - p(S')|(\bar{\mu}_b(S\triangle S') + \mathcal{O}(\epsilon)) + \sum_{b\in B'_G} (\bar{\mu}_b(S\triangle S') + \mathcal{O}(\epsilon))^2$$

$$\leq 2\sqrt{\sum_{b\in B'_G} (\bar{\mu}_b(S') - p(S'))^2}\sqrt{\sum_{b\in B'_G} (\bar{\mu}_b(S\triangle S') + \mathcal{O}(\epsilon))^2} + \sum_{b\in B'_G} (\bar{\mu}_b(S\triangle S') + \mathcal{O}(\epsilon))^2,$$

here the last inequality follows from Cauchy-Schwarz inequality. Next, we bound the last terms in the above expression.

$$\sum_{b\in B'_G} (\bar{\mu}_b(S\triangle S') + \mathcal{O}(\epsilon))^2 \leq \sum_{b\in B'_G} (\bar{\mu}_b(S\triangle S') + \mathcal{O}(\epsilon))(1 + \mathcal{O}(\epsilon))$$

$$\leq 2 \cdot \sum_{b\in B'_G} (\bar{\mu}_b(S\triangle S') + \mathcal{O}(\epsilon))$$

$$\leq 2 \cdot \left(|B'_G|\mathcal{O}(\epsilon) + \sum_{b\in B_G} (\bar{\mu}_b(S\triangle S'))\right)$$

$$\leq 2|B'_G|\left(\mathcal{O}(\epsilon) + \frac{|B_G|}{|B'_G|}\bar{p}_{B_G}(S\triangle S')\right)$$

$$\leq |B'_G|\mathcal{O}(\epsilon).$$

Also, from the property (ii) for $S'$ implies

$$\sum_{b\in B'_G} (\bar{\mu}_b(S') - p(S'))^2 \leq |B'_G|\mathrm{V}(p(S')) + |B'_G|\mathcal{O}\left(\frac{\beta\ln(\frac{1}{\beta})}{n}\right)$$

$$\leq |B'_G|\mathcal{O}\left(\frac{1}{n}\right),$$

here we used equation (3), that implies $V(\cdot) \leq 1/4n$, and $\beta\ln(1/\beta) = \mathcal{O}(1)$. Combining the above three equations we get

$$\sum_{b\in B'_G} (\bar{\mu}_b(S) - p(S))^2 - \sum_{b\in B'_G} (\bar{\mu}_b(S') - p(S'))^2$$

$$\leq 2\sqrt{|B'_G|\mathcal{O}\left(\frac{1}{n}\right)}\sqrt{|B'_G|\mathcal{O}(\epsilon)} + |B'_G|\mathcal{O}(\epsilon) < |B'_G|\mathcal{O}\left(\sqrt{\frac{\epsilon}{n}}\right).$$

Similarly, one can prove the other direction

$$\sum_{b\in B'_G} (\bar{\mu}_b(S') - p(S'))^2 - \sum_{b\in B'_G} (\bar{\mu}_b(S) - p(S))^2 < |B'_G|\mathcal{O}\left(\sqrt{\frac{\epsilon}{n}}\right).$$

Combining the two equations gives

$$\left|\sum_{b\in B'_G} (\bar{\mu}_b(S) - p(S))^2 - \sum_{b\in B'_G} (\bar{\mu}_b(S') - p(S'))^2\right| < |B'_G|\mathcal{O}\left(\sqrt{\frac{\epsilon}{n}}\right).$$

And from equation ([3](#)) we get

$$|\mathrm{V}(p(S)) - \mathrm{V}(p(S'))| \leq \frac{|p(S) - p(S')|}{n} \leq \frac{|p(S \triangle S')|}{n} \leq \mathcal{O}\left(\frac{\epsilon}{n}\right).$$

Combining the above two equations we get

$$\left| \frac{1}{|B'_G|} \sum_{b \in B'_G} (\bar{\mu}_b(S) - p(S))^2 - \mathrm{V}(p(S)) \right|$$

$$\leq \left| \frac{1}{|B'_G|} \sum_{b \in B'_G} (\bar{\mu}_b(S') - p(S'))^2 - \mathrm{V}(p(S')) \right| + \mathcal{O}\left(\sqrt{\frac{\epsilon}{n}}\right) + \mathcal{O}\left(\frac{\epsilon}{n}\right)$$

$$\overset{(a)}{\leq} \mathcal{O}\left(\frac{\beta \ln(\frac{1}{\beta})}{n}\right) + \mathcal{O}\left(\sqrt{\frac{\epsilon}{n}}\right) + \mathcal{O}\left(\frac{\epsilon}{n}\right)$$

$$\overset{(b)}{\leq} \mathcal{O}\left(\frac{\beta \ln(\frac{1}{\beta})}{n}\right), \tag{9}$$

here inequality (a) uses that the property **(ii)** holds for $S'$, (b) uses $\epsilon = \mathcal{O}\left(\frac{\beta^2}{n}\right)$.

This completes the proof of the extension of property **(ii)** to subset $S$ and in a similar fashion property **(v)** can be extended.

Next, we extend property **(iii)** to subset $S$.

Note that

$$\left| \{ b \in B_G : |\bar{\mu}_b(S) - p(S)| \geq t \} \right|$$

$$\overset{(a)}{\leq} \left| \{ b \in B_G : |\bar{\mu}_b(S') - p(S')| + \bar{\mu}_b(S \triangle S') + \mathcal{O}(\epsilon) \geq t \} \right|$$

$$\leq \left| \left\{ b \in B_G : |\bar{\mu}_b(S') - p(S')| \geq \frac{2}{3} \cdot t \right\} \right| + \left| \left\{ b \in B_G : \bar{\mu}_b(S \triangle S') \geq \frac{t}{3} - \mathcal{O}(\epsilon) \right\} \right|$$

$$\leq \left| \left\{ b \in B_G : |\bar{\mu}_b(S') - p(S')| \geq \frac{2}{3} \cdot t \right\} \right| + \frac{\sum_{b \in B_G} \bar{\mu}_b(S \triangle S')}{\frac{t}{3} - \mathcal{O}(\epsilon)}$$

$$\leq \left| \left\{ b \in B_G : |\bar{\mu}_b(S') - p(S')| \geq \frac{2}{3} \cdot t \right\} \right| + |B_G| \frac{\bar{p}_{B_G}(S \triangle S')}{\frac{t}{3} - \mathcal{O}(\epsilon)}$$

$$\leq \left| \left\{ b \in B_G : |\bar{\mu}_b(S') - p(S')| \geq \frac{2}{3} \cdot t \right\} \right| + |B_G| \frac{\mathcal{O}(\epsilon)}{\frac{t}{3} - \mathcal{O}(\epsilon)}.$$

here inequality (a) uses ([8](#)).

Choosing $t = \mathcal{O}\left(\sqrt{\frac{\ln(1/\beta)}{n}}\right)$ in the above equation and putting $\epsilon = \mathcal{O}(\beta^2/n)$ gives

$$\left| \{ b \in B_G : |\bar{\mu}_b(S) - p(S)| \geq \mathcal{O}\left(\sqrt{\frac{\ln(1/\beta)}{n}}\right) \} \right|$$

$$\leq \left| \{ b \in B_G : |\bar{\mu}_b(S') - p(S')| \geq \mathcal{O}\left(\sqrt{\frac{\ln(1/\beta)}{n}}\right) \} \right| + |B_G| \frac{\mathcal{O}(\beta^2/n)}{\mathcal{O}\left(\sqrt{\ln(1/\beta)/n}\right) - \mathcal{O}(\beta^2/n)}$$

$$\leq \mathcal{O}(\beta)|B_G|. \tag{10}$$

here the last step uses property **(ii)** for $S'$. This extends property **(iii)** to subset $S$. Property **(iv)** can be extended similarly. ∎

*Proof of Lemma 19.* The previous lemma showed that the auxiliary properties hold for all subsets in $\mathcal{G}$. Lemma [26](#) showed that these auxiliary properties implies the filtration properties. Combining the two Lemmas completes the proof of Lemma 19.

# E Remaining proofs

## E.1 Proof of Theorem 5

To prove the above theorem we use the following result.

**Theorem 28** ([ADLS17]). *There is an algorithm which, given any $t$ samples $x_1, x_2, ..., x_s \in \mathbb{R}$, returns an $t$-piecewise degree-$d$ polynomial $p'$ which minimizes $||p' - \bar{p}_s||_{\mathcal{F}_{2td}}$ distance between $p'$ and the empirical distribution $\bar{p}_s$, to within additive error $\gamma$ in time $poly(s, t, d, 1/\gamma)$.*

We note that the $t$-piecewise degree-$d$ polynomial $p'$ returned in the above theorem may not always integrate to 1 and is only an approximate Yatracos minimizer, and hence we can not directly use equation (1).

But there is a simple generalization of this equation in [DL01], which applies even when $p'$ returned in the above theorem doesn't integrate to 1 and is only an approximate Yatracos minimizer.

Recall that $\mathcal{Y}(\mathcal{P})$ is Yatracos class of $\mathcal{P}$. Let $p' \in \mathcal{P}$ be such that $||p' - \bar{p}||_{\mathcal{Y}(\mathcal{P})} = \min_{q \in \mathcal{P}} ||q - \bar{p}||_{\mathcal{Y}(\mathcal{P})} + \gamma$ Then [DL01] (exercise 6.2) implies that

$$||p - p'||_{TV} \leq 5 \cdot \text{opt}_{\mathcal{P}}(p) + 4||p - \bar{p}||_{\mathcal{Y}(\mathcal{P})} + 5\gamma.$$

Recall that Yatracos class of $t$-piecewise degree $d$ polynomials, (including those that don't integrate to 1), is $\mathcal{F}_{2td}$.

Theorem 2 provides a polynomial time algorithm that returns a sub-collection $B^* \subseteq B$ of batches whose empirical distribution $\bar{p}_{B^*}$ is close to $p$ in $\mathcal{F}_{2td}$-distance. Then running the algorithm in Theorem 28 for samples in $\bar{p}_{B^*}$ returns a $t$-piecewise degree-$d$ polynomial $p^*$. Then the above equation implies that $p^*$ approximates $p$ in TV distance, to complete the proof of the theorem.

## E.2 Proof of Lemma 6

*Proof.* For two distributions $p$ and $q$ over $\Omega \times \{0, 1\}$, the largest difference between the loss of any classifier $h \in \mathcal{H}$ is related to their $\mathcal{F}_{\mathcal{H}}$-distance,

$$
\begin{aligned}
\sup_{h \in \mathcal{H}} |r_p(h) - r_q(h)| &= \sup_{h \in \mathcal{H}} |\Pr_{(X,Y) \sim p}[h(X) \neq Y] - \Pr_{(X,Y) \sim q}[h(X) \neq Y]| \\
&\leq \sup_{h \in \mathcal{H}} \sum_{y \in \{0,1\}} |\Pr_{(X,Y) \sim p}(h(X) = \bar{y}, Y = y) - \Pr_{(X,Y) \sim q}(h(X) = \bar{y}, Y = y)| \\
&\leq 2||p - q||_{\mathcal{F}_{\mathcal{H}}}.
\end{aligned}
\tag{11}
$$

Then,

$$
\begin{aligned}
&r_p(h^{\text{opt}}(q)) - r_p^{\text{opt}}(\mathcal{H}) \\
&= r_p(h^{\text{opt}}(q)) - r_p(h^{\text{opt}}(p)) \\
&= r_p(h^{\text{opt}}(q)) - r_q(h^{\text{opt}}(q)) + r_q(h^{\text{opt}}(q)) - r_q(h^{\text{opt}}(p)) + r_q(h^{\text{opt}}(p)) - r_p(h^{\text{opt}}(p)) \\
&\leq r_q(h^{\text{opt}}(q)) - r_q(h^{\text{opt}}(p)) + 2 \sup_{h \in \mathcal{H}} |r_q(h) - r_p(h)| \\
&\leq 2 \sup_{h \in \mathcal{H}} |r_q(h) - r_p(h)| \\
&\leq 4||p - q||_{\mathcal{F}_{\mathcal{H}}},
\end{aligned}
$$

here the last inequality uses (11). ∎

## E.3 Proof of Theorem 8

*Proof.* Let $\mathcal{H} : \Omega \to \{0, 1\}$ of Boolean functions with VC dimension $\mathcal{V}_{\mathcal{H}} \geq 1$. And let $(X, Y) \sim p$, where $X \in \Omega$ and $Y \in \{0, 1\}$.

Since $\mathcal{V}_{\mathcal{H}} \geq 1$, then there is at-least one $\omega^* \in \Omega$ and $h_1, h_2 \in \mathcal{H}$, s.t. $h_1(\omega^*) \neq h_2(\omega^*)$, w.l.o.g., let $h_1(\omega^*) = 1$ and $h_2(\omega^*) = 0$.

Next, we define two distributions $p_1$ and $p_2$. Let $\gamma = c\frac{\beta}{\sqrt{n}}$, for some small enough constant $c > 0$ to be chosen later. Let $p_1(\omega^*, 1) = p_2(\omega^*, 0) = \frac{1}{2} + \gamma$, and $p_1(\omega^*, 0) = p_2(\omega^*, 1) = \frac{1}{2} - \gamma$. Both $p_1$ and $p_2$ assigns zero probability to all other points in $\Omega \times \{0, 1\}$.

It is easy to see that, for distribution $p_1$, hypothesis $h_1$ achieves the optimal loss $\frac{1}{2} - \gamma$ and similarly for distribution $p_2$, hypothesis $h_2$ achieves the optimal loss $\frac{1}{2} - \gamma$.

Next, note that for distribution $p_1$ the loss of any classifier $f : \Omega \to \{0, 1\}$ is

$$\Pr_{(X,Y) \sim p_1} (f(\omega^*) \neq Y) = \Pr(f(\omega^*) = 1) \times (\frac{1}{2} - \gamma) + \Pr(f(\omega^*) = 0) \times (\frac{1}{2} + \gamma).$$

Similarly its loss for distribution $p_2$ is

$$\Pr_{(X,Y) \sim p_2} (f(\omega^*) \neq Y) = \Pr(f(\omega^*) = 1) \times (\frac{1}{2} + \gamma) + \Pr(f(\omega^*) = 0) \times (\frac{1}{2} - \gamma).$$

Adding the two losses we get

$$\Pr_{(X,Y) \sim p_1} (f(\omega^*) \neq Y) + \Pr_{(X,Y) \sim p_2} (f(\omega^*) \neq Y) = 1$$

Therefore, every classifier incurs a loss of $\geq 1/2$ for at least one of the two distributions. Since the optimal loss for both distributions is $1/2 - \gamma$, any classifier incurs an excess loss of $\gamma$ for at least one of the distributions among $p_1$ and $p_2$.

The distribution $p$ of the data $(X, Y)$, is chosen to be one of the two distributions $p_1$ and $p_2$ each with probability $1/2$. Then we show that depending on which distribution is chosen as $p$, the adversary can choose its batches such that, even with infinitely many batches, the two distributions are indistinguishable. Therefore, any classifier incurs an excess loss of $\gamma$ with probability $\geq 1/2$.

Note that for every batch, the number of $Y = 1$'s is a sufficient statistic for determining weather $p$ is $p_1$ or $p_2$, and it is distributed either $B(n, \frac{1}{2} + \gamma)$ or $B(n, \frac{1}{2} - \gamma)$. From equation 2.15 in [AJ06], for any $c < 1/12$ and $\gamma = c\beta/\sqrt{n}$, the total variation distance between $B(n, \frac{1}{2} + \gamma)$ or $B(n, \frac{1}{2} - \gamma)$ is $\leq 2\beta$.

Therefore, the adversary can choose distributions $q_1$ and $q_2$, over the number of $Y = 1$'s in the adversarial batches, such that

$$(1 - \beta)B(n, \frac{1}{2} + \gamma) + \beta q_1 = (1 - \beta)B(n, \frac{1}{2} - \gamma) + \beta q_2.$$

Hence, if the good batches are distributed as $B(n, \frac{1}{2} + \gamma)$ then adversary chooses $q_1$ as distribution of the adversarial batches and if good batches are distributed as $B(n, \frac{1}{2} - \gamma)$ then adversary chooses $q_2$ and in both the cases the resultant joint distribution of all the batches is same. Hence the two cases are indistinguishable. ∎

The theorem implies that even with access to infinitely many batches, even for the simplest of the hypothesis class, no algorithm can avoid an excess loss $\Omega(\beta/\sqrt{n})$ with probability $1/2$.

### E.4  Proof of Theorem 9

*Proof.* To prove the theorem, we show how to use algorithm in Theorem 2 that gives "cleaner" batches for $\mathcal{F}_k$-distance, to get "cleaner" batches for $\mathcal{F}_{\mathcal{H}_k}$-distance.

Recall that
$$\mathcal{F}_{\mathcal{H}_k} = \{(\{x \in \mathbb{R} : h(x) = y\}, \bar{y}) : h \in \mathcal{H}_k, y \in \{0, 1\}\}.$$

First divide the collection of sets $\mathcal{F}_{\mathcal{H}_k}$ into two parts: $\mathcal{F}_{\mathcal{H}_k}^0 := \{(\{x \in \mathbb{R} : h(x) = 0\}, 1) : h \in \mathcal{H}_k\}$ and $\mathcal{F}_{\mathcal{H}_k}^1 := \{(\{x \in \mathbb{R} : h(x) = 1\}, 0) : h \in \mathcal{H}_k\}$. Note that $\mathcal{F}_{\mathcal{H}_k} = \mathcal{F}_{\mathcal{H}_k}^0 \cup \mathcal{F}_{\mathcal{H}_k}^1$. Then, from the definition of $\mathcal{F}$ distance, it follows

$$||p - q||_{\mathcal{F}_{\mathcal{H}_k}} = \max\{||p - q||_{\mathcal{F}_{\mathcal{H}_k}^0}, ||p - q||_{\mathcal{F}_{\mathcal{H}_k}^1}\}$$

Hence, it suffices to estimate $p$ in both $\mathcal{F}_{\mathcal{H}_k}^0$ and $\mathcal{F}_{\mathcal{H}_k}^1$ distances.

Since decision regions for each hypothesis $h \in \mathcal{H}_k$, consists of at most $k$-intervals, these collections can be rewritten as $\mathcal{F}^0_{\mathcal{H}_k} := \{(S, 0) : S \in \mathcal{F}_k\}$ and $\mathcal{F}^1_{\mathcal{H}_k} := \{(S, 1) : S \in \mathcal{F}_k\}$.

To learn in $\mathcal{F}^0_{\mathcal{H}_k}$ distance, w.l.o.g., we can remap all points of the form $(x, 1)$ to $(\infty, 0)$. Then this problem is identical to learning in $\mathcal{F}_k$ distance as $y = 0$ is the same for all samples after remapping. Similarly to learn in $\mathcal{F}^1_{\mathcal{H}_k}$ distance we remap all points of the form $(x, 0)$ to $(\infty, 1)$.

Then use the algorithm in Theorem 2 to first remove the adversarial batches for $\mathcal{F}^0_{\mathcal{H}_k}$ distance, and then for the remaining batches again use the same algorithm to remove adversarial batches for $\mathcal{F}^1_{\mathcal{H}_k}$ distance. The empirical distribution $\bar{p}_{B^*}$ of the batches $B^* \subseteq B$ remaining in the end, approximates $p$ in both $\mathcal{F}^0_{\mathcal{H}_k}$ and $\mathcal{F}^1_{\mathcal{H}_k}$ distances to an accuracy $\mathcal{O}(\Delta)$. Therefore, it estimates $p$ in $\mathcal{F}_{\mathcal{H}_k}$ distance to the same accuracy.

Then use the polynomial-time algorithm [Maa94] to find the empirical risk minimizer $h \in \mathcal{H}_k$ for empirical distribution $\bar{p}_{B^*}$. Then Lemma 6 implies that the optimal classifier $h^{\text{opt}}(\bar{p}_{B^*})$ for the empirical distribution $\bar{p}_{B^*}$, of the cleaner batch collection $B^*$, will have a small-excess-classification-loss $\mathcal{O}(\Delta)$ for $p$. This completes the proof of the theorem. ∎