[Reviews · NeurIPS 2020]

Review 1

Summary and Contributions: This paper extends recent algorithmic work for the problem of learning from untrusted batches. The focus is on the setting where the underlying distribution has additional structure, in which case more sample efficient algorithms are possible. This paper develops sample and computationally efficient algorithms for such settings.

Strengths: The paper presents novel theoretical results that are highly relevant for the machine learning community. At a high-level, the paper combines in a natural but non-trivial way VC theory with the filtering framework (developed in prior work).

Weaknesses: One potential limitation is the assumption that each good batch contains samples from the distribution p -- as opposed to some distribution that is close to p in L1 distance. The original framework of Qiao-Valiant covers this setting and some of the previous efficient algorithms work in this setting as well. It is not immediately clear from the writeup if this is an inherent limitation of the methodology proposed in this paper.

Correctness: The proofs appear to be correct. However, the paper is fairly long and *all* the proofs appear in an appendix.

Clarity: This writing is quite clear.

Relation to Prior Work: While the prior work, in terms of theorem statements, is sufficiently explained, very limited space is devoted in explaining the the techniques and in particular providing a comparison to prior techniques. Some specific comments/questions are provided below.

Reproducibility: Yes

Additional Feedback: From the writeup it is not clear how the techniques are related to prior work. For example: 1) The filtering framework of [J019], that the current work builds on, is itself an adaptation of the filtering technique developed in the robust statistics literature, [DKK+16], etc. It would be appropriate to explain this and also elaborate on the differences between [J019] and the current application of filtering (which is done to some extent on p. 8). 2) The idea of using VC theory for computationally efficient density estimation has been used in a sequence of prior works (including ADLS17). The current paper builds on these ideas. It would be useful for the non-expert reader if this was clarified. 3) The paper contains essentially zero proofs in the first 8 pages. I understand there are space issues. But it would have certainly been possible to include 2-3 pages of content and move some of the secondary contributions to the appendix. With this presentation, a non-expert reviewer either needs to believe correctness or read the appendix. ======== Post rebuttal: -- The authors responded that their algorithm can be extended to work for the case that the samples from each good batch come from a distribution that is close to p. And that this follows immediately from their approach. It would be useful if this is included in the final version. -- The authors also promised that they will revise their presentation to address the relation to prior work at a technical level. Given the above, I am happy to increase my score to 7.


Review 2

Summary and Contributions: This paper generalizes the filter algorithm proposed in [JO19] for learning from batches from discrete distribution learning to continuous domain. The main technical lemma shows that the proposed filter algorithm can “clean” the batches to guarantee uniform convergence with near optimal sample complexity. This lemma is leveraged to learn piecewise polynomial distribution, binary classification. Though the result is generally statistical in its nature and does not yield a computation efficient algorithm, the paper leverages an efficient learning algorithm for unions of intervals to give efficient algorithms for learning piecewise polynomial distributions and union of intervals classifiers.

Strengths: This paper generalizes the filter algorithm proposed in [JO19] for learning from batches from discrete distribution learning to continuous domain, and provides the first efficient robust batch learning algorithm for several fundamental learning problems.

Weaknesses: The proof for the sample efficient filter for general VC class follows from a very similar technical approach as in [JO19].

Correctness: Yes

Clarity: yes

Relation to Prior Work: Yes

Reproducibility: Yes

Additional Feedback: I have read the authors' feedback and my score does not change.


Review 3

Summary and Contributions: I thank the authors for their feedback. --- The authors present robust learning algorithms using batches for 1) learning structured distributions 2) classification problems 3) 1d classification problems All above algorithms are based on a common filtering framework.

Strengths: The authors give tractable algorithms that match sample complexity lower bounds up to log factors.

Weaknesses: It's unclear how the authors' bounds will break down if we do not make the assumption that the distribution is structured. Also, it is unclear how large is the polynomial time complexity. If time complexity's polynomial degree is too high, then the algorithms are still impractical. And the authors should also emphasize more how their work is technically superior to [JO19].

Correctness: Yes

Clarity: There are many results in this paper and it'd be nicer if they're more organized.

Relation to Prior Work: Yes

Reproducibility: Yes

Additional Feedback:


Review 4

Summary and Contributions: Sample complexity bound for the task of distribution-learning in the setting of batch sampling with adversarial batches, i.e. a $\beta$ fraction of the batches might get maliciously corrupted. The bounds are almost tight as they match the lower information-theoretic bounds for this case, up to logarithmic factors. The main novelty is that the result can be applied to distributions over infinite or even continues domains. As a corollary the authors provide an efficient learning algorithm, in terms of sample and run-time, for learning piecewise polynomial distributions. The notions of interest are - $\mathcal{F}$-distance, which generalizes TV-distance. - Yatracos class and the Yatracos-minimizer. - VC-dimension and the VC-inequality and convergence theorems.

Strengths: The paper seems to be of high-quality. The results are impressive, non-trivial and interesting, in the sense that they might lead to further research in their direction. The authors did a through work and provided several applications to their main result which are of individual interest. Update: I read the authors' feedback and my score stays the same.

Weaknesses: Didn't find any specific weakness.

Correctness: As far as I saw - it is correct.

Clarity: Yes.

Relation to Prior Work: Yes.

Reproducibility: Yes

Additional Feedback:

[Author Response · NeurIPS 2020]

We thank the reviewers for the valuable time they have invested during this difficult period to review the paper and provide helpful suggestions for improving the manuscript. We also appreciate their complimentary comments, including "The paper presents novel theoretical results that are highly relevant for the machine learning community" (Reviewer 1), "The paper provides the first efficient robust batch learning algorithm for several fundamental learning problems" (Reviewer 2), and "The paper seems to be of high-quality. The results are impressive, non-trivial and interesting" (Reviewer 4). The remainder of this response mostly addresses suggestions and questions raised by Reviewers 1 and 3.

Both Reviewers 1 and 3 ask us to elaborate on the differences between [JO19] and this paper. The differences fall in two categories: technique, and applications. In terms of technique, [JO19] does not leverage the distribution structure, it simply uses all domain subsets as filters. It therefore requires a sample size linear in the domain size, which is prohibitive for large domains and impossible for the all-important infinite and continuous domains.

By contrast, this paper utilizes the distribution's structure, or even rough proximity to a structure, to identify a much smaller class of filters that as we show, suffices to address adversarial batches. This significant improvement, that as Reviewer 1 writes, requires a "non-trivial" combination of VC theory and the filtering framework, allows us to remove the sample complexity's dependence on the domain size and to greatly extend the reach of the filtering algorithm. We then apply it to derive (1) robust estimation for whole range of distributions, including infinite and even continuous, that hitherto could not be learned robustly, (2) robustness results for vital learning tasks, most notably classification, that were not addressed in [JO19]. Note also that: (1) our information theoretic results on classification assume only that the classifier's hypothesis has a finite VC dimension – the most common assumption in learning theory, and (2) our efficient classification algorithm applies to the fundamental problem of 1-d interval classifiers.

Reviewers 1 and 3 also ask related questions about how the results will hold if the distributions are unstructured (Reviewer 3) or may vary by a small amount from each other (Reviewer 1). If the distribution is completely unstructured, then as pointed out in lines 49-53 of the paper, the sample complexity grows linearly with the domain size, hence one cannot learn the type of distributions addressed in this paper. If the distribution can be approximated by a structured distribution then it is covered by the "opt density estimation framework" utilized in the paper, see e.g., lines 194-196.

Regarding Reviewer 1's specific question whether the technique also applies when the distributions underlying genuine batches differ from a common target distribution by a small TV distance, say $\eta > 0$. For simplicity, we presented the analysis for $\eta = 0$, but as noted in [JO19] for unstructured distributions, the filtering technique easily adapts to $\eta > 0$. For example, in density estimation the trivial empirical estimator achieves $\mathcal{O}(\eta + \beta)$ TV-error, or $\mathcal{O}(\beta)$ when $\eta = 0$. Even for binary alphabets, the lower bound is $\Omega(\eta + \beta/\sqrt{n})$, hence no algorithm can reduce the effect of the disparity between the batches and target distributions. Filtering reduces the effect of adversarial batches from $\mathcal{O}(\beta)$ to $\tilde{\mathcal{O}}(\beta/\sqrt{n})$. Since we cannot do anything sophisticated about $\eta$, the proof and algorithm easily extend to $\eta > 0$. For this reason we presented the simplest problem that captures the essence of the technique. We will add a similar explanation to the final version.

Reviewer 1 suggests that we elaborate on the relationship between filtering methods for Gaussian mean estimation derived e.g., in [DKK+16], and [JO19]. This relation was explained in [JO19]. Section 3 of this paper, mentions the many important contributions of [DKK+16, DKK+17, SCV17], the recent survey [DKK+19], and others, but for brevity does not repeat the explanation in [JO19]. To enhance the reader's understanding of the context, in the final version of the paper we will follow the reviewer's advice and expand this discussion and elaborate on the specific relation to [JO19].

Reviewer 1 similarly suggests that we elaborate on previous use of VC theory in structured density estimation (including [ADLS17]). Please note that Section 3 of the paper starts by stating that "The current results extend several long lines of work on estimating structured distributions, including [O'B16, Dia16, AM18, ADLS17]" and that we provide specific references to [ADLS17] in three additional locations in the main paper and several more times in the appendix. Also note that the previous applications of VC theory were for non-robust learning, hence somewhat different from the current application that requires several new ideas. For the reader's benefit we will follow the reviewer's advice and elaborate on the use of VC theory in density estimation in non-robust setting.

Reviewer 1 also suggested that we move some of the applications from the main paper to the appendix and some proofs from the appendix to the main paper. We fully sympathize with the reviewer's desire to see more hard proofs in the paper itself, but felt that one of the paper's main contributions is showing broad audiences that adversarial batches can be addressed efficiently for a large class of practical problems. We also note that Reviewer 4's response to question 2 seems to appreciate this information. We will try to accommodate Reviewer 1's request by including as much information about the proofs as we can in the extra page of the papers' final version.

Finally, Reviewer 3 asks about the time complexity of the paper's two efficient algorithms: learning piecewise polynomials, and interval classification. Both algorithms have very reasonable complexities. Learning $t$-piecewise, degree-$d$ polynomial distributions takes $\mathcal{O}(m \cdot n^2(1 + t \cdot d \cdot \beta/\sqrt{n}))$ time, and $t$-interval classification takes $\mathcal{O}(m \cdot n^2(1 + t \cdot \beta/\sqrt{n}))$ time. Since there is a total of $m \cdot n$ samples, these complexities are not too high. We will mention these time complexities explicitly in the paper.

[Meta-Review · NeurIPS 2020]

This paper addresses the question of learning structured distributions from batches when a constant fraction of the batches might be corrupted. This problem has been of considerable recent interest. This paper studies the setting where the underlying distribution has additional structure (namely, piece polynomial density function), in which case more sample efficient algorithms are possible. This paper develops sample and computationally efficient algorithms for such settings. The reviewers were convinced that this paper makes important technical contributions in extending recent work on this problem to the structured setting. I recommend accepting this paper.